# MQ-VAE: TRAINING VECTOR-QUANTIZED NETWORKS VIA META LEARNING

## ABSTRACT

Deep neural networks with discrete latent variables are particularly well-suited for tasks that naturally involve sequences of discrete symbols. The vector-quantized variational auto-encoder (VQ-VAE) has made significant progress in this area by leveraging vector quantization. However, while much effort has been put into maximizing codebook utilization, this does not always result in better performance. Additional challenges include quantization errors in the VQ layer and the lack of direct integration of task loss into the codebook objective. To address these issues, we propose Meta-Quantized Variational Auto-Encoder (MQ-VAE), a bi-level optimization-based vector quantization framework inspired by meta-learning. In MQ-VAE, the codebook and encoder-decoder pair are optimized at different levels, with the codebook treated as hyperparameters optimized via hyper-gradient descent. This approach effectively tackles these challenges within a unified framework. The evaluation of MQ-VAE on two computer vision tasks demonstrates its superiority over existing methods and ablation baselines. Code is available at https://anonymous.4open.science/r/MQVAE-B52C.

## 1 INTRODUCTION

Learning discrete latent variables is often preferable for tasks that are naturally modeled as sequences of discrete symbols, such as language and speech (Vinyals et al., 2015). Vector-quantized networks (VQNs) are a type of network that learns latent variables via vector quantization (VQ, Gray (1984)), which quantizes features into clusters known as codewords. VQNs were first introduced as the vector-quantized variational auto-encoder (VQ-VAE, Van Den Oord et al. (2017)) in the context of generative models. Later studies demonstrated that training an autoregressive prior on discrete representations learned through vector quantization leads to powerful image generation models (Razavi et al., 2019; Roy et al., 2018; Ramesh et al., 2021; Esser et al., 2021; Chang et al., 2023). VQNs have also yielded impressive results in speech generation (Dhariwal et al., 2020) and other areas beyond generation, such as image representation learning (Caron et al., 2020) and speech representation learning (Chung et al., 2020).

The traditional way of training VQNs, as used in VQ-VAE, is to learn a codebook $\mathcal{C}$, whose elements, known as codewords, provide a compressed semantic representation of the input data. The embedding of the encoded data from the encoder $F_\phi$ then goes through the quantization operation by selecting its nearest neighbor in $\mathcal{C}$. The selected codeword replaces the embedding and is passed to the decoder $G_\theta$ for an output. Due to the non-differentiability of the quantization operation, a straight-through estimator (STE, Bengio et al. (2013)) is usually used to enable gradient flow through the VQ layer to the encoder when backpropagating. Since the backpropagating gradient bypasses the codebook in STE, the codebook is instead optimized using a vector quantization objective that brings the embedding and selected codewords together. In this way, the entire model is optimized in an end-to-end manner.

However, several challenges exist in the previous training framework. First, vector quantization performs poorly when the number of actively used codes is small, a problem known as 'index collapse' (Kaiser et al., 2018). This is mainly because of the sparsity of the gradient, meaning only the selected codewords are updated. Existing works such as Lee et al. (2022), Kaiser et al. (2018), and Yu et al. (2021) have explored explicitly encouraging high code utilization rates. However, the implicit assumption that a higher code utilization rate necessarily leads to better performance is not

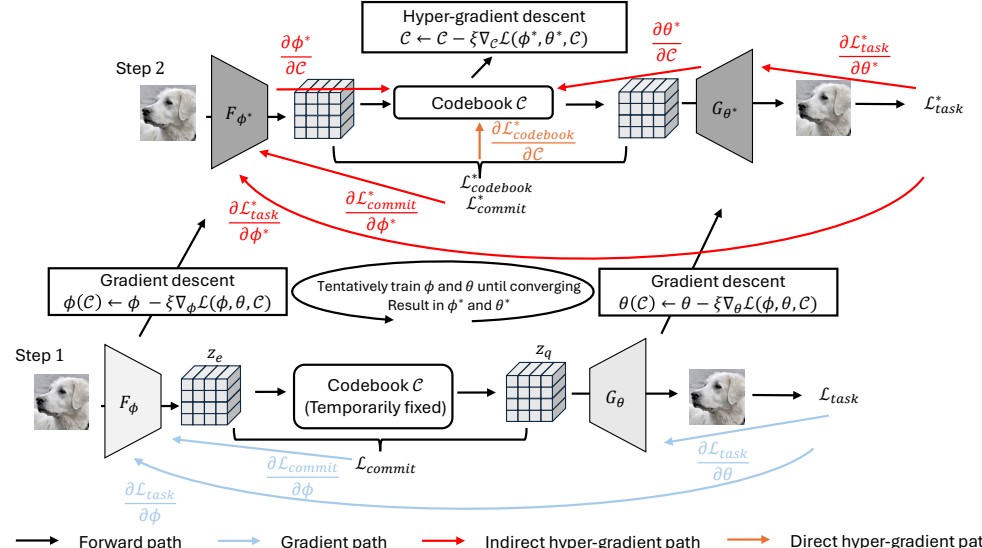

Figure 2: **Gradient paths in MQ-VAE.** The lower level is optimized by gradient descent. The upper level is optimized by hyper-gradient descent with direct hyper-gradient and indirect hyper-gradient. Here, we use $\mathcal{L}^*_{\text{task}}$, $\mathcal{L}^*_{\text{commit}}$ and $\mathcal{L}^*_{\text{codebook}}$ to denote the corresponding loss computed with $\phi^*(\mathcal{C})$ and $\theta^*(\mathcal{C})$.

always valid (Huh et al., 2023; Zheng & Vedaldi, 2023)—a high utilization rate in tasks that have more codes than necessary results in redundancy, which may even lead to overfitting. Second, the quantization error at the VQ layer can introduce a gradient estimation gap when using STE, making the training of the encoder and decoder biased and unstable. Third, the codebook is solely optimized with the vector quantization objective, which only focuses on aligning the embedding distribution and codeword distribution. Since the gradient from task loss bypasses the codebook in STE when backpropagating, the update of the codebook is task-unaware, which potentially compromises its optimality.

Drawing inspiration from meta-learning (Finn et al., 2017), we propose a simple and unified framework called **M**eta-**Q**uantized **V**ariational **A**uto-**E**ncoder (MQ-VAE) to address the challenges mentioned above. Built directly on the traditional vector quantization framework with the same network architecture, we introduce an asymmetric bi-level optimization problem, where the codebook acts as hyperparameters and the encoder-decoder pair as parameters. As shown in Figure 1, the codebook and the encoder-decoder pair are optimized at the upper and lower levels, respectively. At the upper level, the codebook anticipates the future performance of the encoder-decoder pair by tentatively training them until convergence (or unrolling for several steps as a surrogate) with the codebook temporarily fixed. Then, the codebook is optimized to minimize the loss via hyper-gradient descent for one step using the optimal encoder and decoder, which are functions of the codebook. At the lower level, the

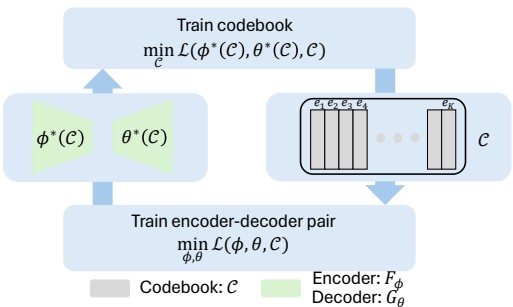

Figure 1: **Bi-level optimization in MQ-VAE.** MQ-VAE learns the codebook and encoder-decoder pair using a bi-level optimization framework. At the lower level, the encoder-decoder pair is trained to converge while keeping the codebook fixed. At the upper level, the codebook is optimized via hyper-gradient descent using the optimal encoder-decoder pair.

tentative training steps of the encoder-decoder pair in the first step are undone since a better codebook has been found by hyper-gradient descent. Instead, the encoder and decoder are optimized for one step by gradient descent using the updated codebook. The two levels of optimization are performed iteratively until convergence.

The three challenges above can be effectively addressed within our cohesive framework. MQ-VAE resembles online K-means, where the codebook acts as the K-means cluster centroids. When the centroids are adjusted so that the distribution of codewords and embedding perfectly match, zero quantization error is achieved, and no gradient estimation gap occurs in STE. Intuitively, our method employs an alternated optimization structure in which the codebook at the upper level reduces quantization error via the vector quantization objective before each update at the lower level, promoting more stable training of the encoder and decoder. Notably, our approach goes beyond merely considering quantization error as a determinant of the encoder-decoder pair's training quality. Hypergradient descent explicitly integrates the subsequent performance of the encoder and decoder into consideration. In doing so, MQ-VAE learns a superior codebook that improves encoder-decoder training in the long run without fully relying on heuristic assumptions about codebook usage or quantization error. Interestingly, hyper-gradient descent introduces another benefit: unlike the traditional codebook strategy where task loss has no impact on the codebook, the gradient from the task loss can now reach the codebook through various paths (See Figure 2). This makes the codebook task-aware and improves overall performance. Furthermore, in our framework, a higher proportion of codewords are updated each step. This is because different sets of codewords are expected to be selected during each iteration of the tentative encoder-decoder training, and all selected codewords are updated at the final step, which potentially mitigates the gradient sparsity issue.

Our work makes the following key contributions:

- We propose MQ-VAE, an innovative vector quantization framework based on bi-level optimization. Unlike the traditional codebook optimization procedure, our approach optimizes the codebook in a meta-learning fashion via hyper-gradient descent. MQ-VAE can learn a superior codebook that improves encoder-decoder training in the long run without fully relying on heuristic assumptions about codebook usage or quantization error.

- A detailed analysis of the hyper-gradient reveals how MQ-VAE can enhance the gradient guidance during codebook training. Additionally, we show that in our algorithm, the task loss can directly affect the codebook, which was largely ignored in previous works.

- We demonstrate the superiority of MQ-VAE with two computer vision tasks. MQ-VAE significantly outperforms several comparison baselines and ablation methods, highlighting its effectiveness.

## 2 RELATED WORK

### 2.1 VECTOR-QUANTIZED NETWORKS

The vector quantization networks (VQNs) were first introduced as a generative model named vector-quantized variational auto-encoder (VQ-VAE), which maps continuous embedding to discrete codewords using a learned codebook by vector quantization (VQ). Straight-through estimator (STE) (Bengio et al., 2013) is applied in the original framework to address the non-differentiability of the VQ layer. However, this simple approximation causes problems, as discussed in the introduction, and significant efforts have been made to improve the VQ layer. Łańcucki et al. (2020), Zeghidour et al. (2021), and Dhariwal et al. (2020) propose resetting codewords that are not selected for certain iterations to increase codebook utilization. Kaiser et al. (2018), Guo et al. (2024), and Yu et al. (2021) suggest projecting both embedding and codewords into subspaces and applying vector quantization to each subspace. In this way, the dimensionality of codewords is split into several groups, and a higher codebook utilization is achieved by conducting quantization in each group separately. Gumbel-VQ, a variation proposed in a public repository by Karpathy (2021) and used as a comparison method in Huh et al. (2023), provides a continuous approximation of vector quantization. By doing so, the bottleneck is differentiable, and the codebook can be trained using standard backpropagation. Other related works include affine reparameterization (Huh et al., 2023), $l_2$ normalization (Yu et al., 2021), and probabilistic reformulations (Roy et al., 2018; Takida et al., 2022).

### 2.2 BI-LEVEL OPTIMIZATION

Bi-level optimization (BLO) has been widely applied in various machine learning tasks, with meta-learning (Finn et al., 2017; Rajeswaran et al., 2019) being one of the most prominent applications.

Other applications include neural architecture search (NAS, Liu et al. (2018); Zhang et al. (2021)) and hyperparameter optimization (HPO, Lorraine et al. (2020); Franceschi et al. (2017)). Despite its widespread use, solving BLO problems can be challenging due to the inherently nested nature of two optimization tasks. Gradient-based BLO (Choe et al., 2023) has received much attention because it can scale to high-dimensional problems with a large number of trainable parameters.

In this work, we extend the application of gradient-based BLO to develop a novel codebook training approach for the vector quantization framework. In the spirit of meta-learning, MQ-VAE treats the codebook as hyperparameters with the objective of improving the training of the encoder and decoder. The effectiveness of the codebook is validated and updated based on the performance potentiality of the encoder and decoder, analogous to validating the effectiveness of model initialization in meta-learning. A similar strategy of hyper-gradient descent in our work is also used in the meta-learning literature.

## 3 PRELIMINARY

A vector-quantized network (VQN) is a neural network consisting of a vector-quantization (VQ) layer $h_{\mathcal{C}}(\cdot)$:

$$\mathbf{y} = G_\theta(h_{\mathcal{C}}(F_\phi(\mathbf{x}))) = G_\theta(h_{\mathcal{C}}(\mathbf{z}_e)) = G_\theta(\mathbf{z}_q) \tag{1}$$

Here, $\mathbf{z}_e$ denotes the embedding obtained by applying the encoder $F_\phi$ (parameterized by $\phi$) to the input $\mathbf{x}$. $\mathbf{z}_q$ denotes the quantized embedding obtained by applying the VQ layer $h_{\mathcal{C}}$ to $\mathbf{z}_e$. $\mathbf{y}$ denotes the output of the decoder $G_\theta$ (parameterized by $\theta$), which takes $\mathbf{z}_q$ as input. The VQ layer $h_{\mathcal{C}}(\cdot)$ quantizes $\mathbf{z}_e$ by selecting a vector from the codebook $\mathcal{C} = \{\mathbf{e}_1, \mathbf{e}_2, \dots, \mathbf{e}_k\}$ based on a distance measure $d(\cdot, \cdot)$:

$$\mathbf{z}_q = \mathbf{e}_k, \quad \text{where} \quad k = \arg\min_j d(\mathbf{z}_e, \mathbf{e}_j) \tag{2}$$

Here, a stored vector $\mathbf{e}_i$ is referred to as the codeword, and the index $i$ as the code. Euclidean distance is the standard distance measure for $d(\cdot, \cdot)$ (Van Den Oord et al., 2017). Note that the quantized embedding $\mathbf{z}_q$ is a subset of $\mathcal{C}$, and updating $\mathbf{z}_q$ corresponds to partially updating $\mathcal{C}$.

The task loss $\mathcal{L}_{\text{task}}(G_\theta(h_{\mathcal{C}}(F_\phi(\mathbf{x}))), \mathbf{y})$ is not continuously differentiable due to the $\arg\min$ operator in $h_{\mathcal{C}}$. To address this, a straight-through estimator (STE, Bengio et al. (2013)) is applied with the non-differentiable part $\frac{\partial \mathbf{z}_q}{\partial \mathbf{z}_e}$ ignored:

$$\frac{\partial \mathcal{L}_{\text{task}}}{\partial \phi} \approx \frac{\partial \mathcal{L}_{\text{task}}}{\partial \mathbf{y}} \frac{\partial \mathbf{y}}{\partial \mathbf{z}_q} \frac{\partial \mathbf{z}_e}{\partial \phi} \tag{3}$$

To ensure an accurate STE, $\mathbf{z}_e$ and $\mathbf{z}_q$ are aligned using two additional losses:

$$\mathcal{L}_{\text{commit}}(\mathbf{z}_q, \mathbf{z}_e) = d(\mathbf{z}_e, \text{sg}[\mathbf{z}_q]) \tag{4}$$

$$\mathcal{L}_{\text{codebook}}(\mathbf{z}_q, \mathbf{z}_e) = d(\text{sg}[\mathbf{z}_e], \mathbf{z}_q) \tag{5}$$

Here, sg denotes the stop gradient operator, which treats the entire term as a constant with zero partial derivatives. The commitment loss $\mathcal{L}_{\text{commit}}$ moves the embedding to the selected codewords, while the codebook loss $\mathcal{L}_{\text{codebook}}$, also known as the vector quantization objective, moves the selected codewords toward the centroids of the embedding. Overall, a differentiable objective is minimized:

$$\min_{\phi, \theta, \mathcal{C}} \mathbb{E}_{(\mathbf{x}, \mathbf{y}) \sim \mathcal{D}}[\mathcal{L}_{\text{task}}(G_\theta(h_{\mathcal{C}}(F_\phi(\mathbf{x}))), \mathbf{y})$$

$$+ \beta \cdot \mathcal{L}_{\text{commit}}(h_{\mathcal{C}}(F_\phi(\mathbf{x})), F_\phi(\mathbf{x})) + \mathcal{L}_{\text{codebook}}(h_{\mathcal{C}}(F_\phi(\mathbf{x})), F_\phi(\mathbf{x}))] \tag{6}$$

where $\beta$ is a scalar that balances the importance of updating $\mathbf{z}_e$ and $\mathbf{z}_q$. In this training framework, the decoder optimizes the first loss term, the encoder optimizes the first and the middle loss terms, and the codebook is optimized only by the last loss term.

## 4 METHODOLOGY

### 4.1 OVERVIEW OF MQ-VAE

We propose to optimize the codebook and encoder-decoder pair by solving a bi-level optimization problem. Recall that $h_{\mathcal{C}}$ denotes the VQ layer with the codebook $\mathcal{C}$ as learnable parameters and

---

**Algorithm 1:** MQ-VAE

---

**Input:** Dataset $\mathcal{D}$

1   Initialize $\phi$, $\theta$ and $\mathcal{C}$ for $F_\phi$, $G_\theta$ and $h_\mathcal{C}$

2   **while** *not converged* **do**

3     |   Update $\mathcal{C}$ by descending $\nabla_\mathcal{C}\mathcal{L}(\phi - \xi\nabla_\phi\mathcal{L}(\phi,\theta,\mathcal{C}), \theta - \xi\nabla_\theta\mathcal{L}(\phi,\theta,\mathcal{C}), \mathcal{C})$

4     |   ($\xi = 0$ if using alternated optimization)

5     |   Update $\phi$ and $\theta$ by descending $\nabla_\phi\mathcal{L}(\phi,\theta,\mathcal{C})$ and $\nabla_\theta\mathcal{L}(\phi,\theta,\mathcal{C})$

**Output:** $\phi^*$, $\theta^*$ and $\mathcal{C}^*$

---

$F_\phi$ and $G_\theta$ denote the encoder and decoder with learnable parameters $\phi$ and $\theta$, respectively. In both levels, we consider a loss $\mathcal{L}$ defined on dataset $\mathcal{D}$ in form of Eq. 6, i.e., the sum of three terms: $\mathcal{L}_{\text{task}}$, $\mathcal{L}_{\text{commit}}$, and $\mathcal{L}_{\text{codebook}}$. At the lower level, $F_\phi$ and $G_\theta$ are trained by minimizing $\mathcal{L}(\phi,\theta,\mathcal{C})$. Since the codebook $\mathcal{C}$ is temporarily fixed, the optimal encoder $\phi^*(\mathcal{C})$ and decoder $\theta^*(\mathcal{C})$ are functions of $\mathcal{C}$. At the upper level, we determine the optimal codebook $\mathcal{C}^*$ with $\phi^*(\mathcal{C})$ and $\theta^*(\mathcal{C})$ by minimizing $\mathcal{L}(\phi^*(\mathcal{C}), \theta^*(\mathcal{C}), \mathcal{C})$. This bi-level optimization problem is solved using an efficient gradient-based algorithm, where the two levels are optimized iteratively until convergence. Related convergence analyses of this type of gradient-based bi-level optimization algorithms can be found in Pedregosa (2016), Rajeswaran et al. (2019), and references therein.

In contrast to bi-level optimization literature, where the two levels are usually optimized on distinct datasets, we use a single dataset $\mathcal{D}$ for both levels. In fact, it remains unclear whether the dataset split can provide better performance in general cases (Bai et al., 2021). In this work, we do not do a dataset split for better data utilization.

### 4.2   A Bi-level Optimization Framework

**Lower Level**   At the lower level, we train the encoder $F_\phi$ and decoder $G_\theta$ by minimizing $\mathcal{L}(\phi,\theta,\mathcal{C})$. Specifically, we aim to find the optimal values of $\phi$ and $\theta$ with $\mathcal{C}$ temporarily fixed, resulting in the following optimization problem:

$$\phi^*(\mathcal{C}), \theta^*(\mathcal{C}) = \arg\min_{\phi,\theta} \mathcal{L}(\phi,\theta,\mathcal{C}) \tag{7}$$

Here, $\phi^*(\mathcal{C})$ and $\theta^*(\mathcal{C})$ denote the optimal solutions for $\phi$ and $\theta$, which are functions of $\mathcal{C}$ since the $\arg\min$ operation does not take $\mathcal{C}$ as an argument.

**Upper Level**   At the upper level, the codebook $\mathcal{C}$ is trained by minimizing the loss of the same functional form but using $\phi^*(\mathcal{C})$ and $\theta^*(\mathcal{C})$ that were optimally learned at the lower level as arguments. The loss then only depends on $\mathcal{C}$, and the upper-level optimization problem is formulated as:

$$\min_\mathcal{C} \mathcal{L}(\phi^*(\mathcal{C}), \theta^*(\mathcal{C}), \mathcal{C}) \tag{8}$$

**A Bi-level Optimization Framework**   By integrating the two levels of optimization problems, we present the overall bi-level optimization problem as:

$$\min_\mathcal{C} \mathcal{L}(\phi^*(\mathcal{C}), \theta^*(\mathcal{C}), \mathcal{C})$$
$$s.t. \quad \phi^*(\mathcal{C}), \theta^*(\mathcal{C}) = \arg\min_{\phi,\theta} \mathcal{L}(\phi,\theta,\mathcal{C}) \tag{9}$$

Note that these two levels of optimization problems are mutually dependent on each other. The solution to the optimization problem at the lower level, $\phi^*(\mathcal{C})$ and $\theta^*(\mathcal{C})$ serves as a parameter for the upper-level problem, while non-optimal variable $\mathcal{C}$ at the upper level acts as a parameter for the lower-level problem. By solving the two interconnected problems jointly, we can learn $\phi^*$, $\theta^*$, and $\mathcal{C}^*$ in an end-to-end manner.

**Optimization Algorithm**   We employ a gradient-based optimization algorithm to solve the bi-level optimization problem presented in Eq. 9 iteratively. Gradient descent can be applied directly to the lower-level problem; however, a significant challenge exists at the upper level: precisely

computing the hyper-gradient, i.e., the gradient of the upper-level objective with respect to $\mathcal{C}$, can be computationally prohibitive due to the lack of an analytical solution for $\phi^*(\mathcal{C})$ and $\theta^*(\mathcal{C})$. To address this issue, we use the following one-step approximation, inspired by Finn et al. (2017):

$$\nabla_\mathcal{C}\mathcal{L}(\phi^*(\mathcal{C}), \theta^*(\mathcal{C}), \mathcal{C}) \approx \nabla_\mathcal{C}\mathcal{L}(\phi - \xi\nabla_\phi\mathcal{L}(\phi, \theta, \mathcal{C}), \theta - \xi\nabla_\theta\mathcal{L}(\phi, \theta, \mathcal{C}), \mathcal{C}) \tag{10}$$

where $\xi$ is the learning rate for the lower-level problem. One-step unrolled approximated solutions, $\phi'(\mathcal{C}) = \phi - \xi\nabla_\phi\mathcal{L}(\phi, \theta, \mathcal{C})$ and $\theta'(\mathcal{C}) = \theta - \xi\nabla_\theta\mathcal{L}(\phi, \theta, \mathcal{C})$, are used as surrogates for $\phi^*(\mathcal{C})$ and $\theta^*(\mathcal{C})$. This is equivalent to introducing a surrogate objective $\mathcal{L}(\phi - \xi\nabla_\phi\mathcal{L}(\phi, \theta, \mathcal{C}), \theta - \xi\nabla_\theta\mathcal{L}(\phi, \theta, \mathcal{C}), \mathcal{C})$ that closely resembles the upper-level objective in Eq. 8. In principle, multiple steps can be unrolled to achieve a more accurate approximation.

A straightforward computation of Eq. 10 requires backpropagating through the optimization process at the lower level. Differentiation through gradient descent has been explored in Maclaurin et al. (2015) and can be achieved using automatic differentiation packages without explicit programming. However, when multiple steps are unrolled, the memory and computational burden increase significantly. Therefore, we employ a further approximation by noticing Eq. 10 can be computed using the chain rule followed by a finite difference approximation (Liu et al., 2018) as

$$\nabla_\mathcal{C}\mathcal{L}(\phi - \xi\nabla_\phi\mathcal{L}(\phi, \theta, \mathcal{C}), \theta - \xi\nabla_\theta\mathcal{L}(\phi, \theta, \mathcal{C}), \mathcal{C}) \tag{11}$$

$$=\nabla_\mathcal{C}\mathcal{L}(\phi', \theta', \mathcal{C}) - \xi\nabla^2_{\mathcal{C},\phi}\mathcal{L}(\phi, \theta, \mathcal{C})\nabla_{\phi'}\mathcal{L}(\phi', \theta', \mathcal{C}) - \xi\nabla^2_{\mathcal{C},\theta}\mathcal{L}(\phi, \theta, \mathcal{C})\nabla_{\theta'}\mathcal{L}(\phi', \theta', \mathcal{C}) \tag{12}$$

$$\approx\nabla_\mathcal{C}\mathcal{L}(\phi', \theta', \mathcal{C}) - \xi\frac{\nabla_\mathcal{C}\mathcal{L}(\phi^+, \theta, \mathcal{C}) - \nabla_\mathcal{C}\mathcal{L}(\phi^-, \theta, \mathcal{C})}{2\epsilon} - \xi\frac{\nabla_\mathcal{C}\mathcal{L}(\phi, \theta^+, \mathcal{C}) - \nabla_\mathcal{C}\mathcal{L}(\phi, \theta^-, \mathcal{C})}{2\epsilon}$$
$$\tag{13}$$

where $\phi^\pm = \phi \pm \epsilon\nabla_{\phi'}\mathcal{L}(\phi', \theta', \mathcal{C})$, $\theta^\pm = \theta \pm \epsilon\nabla_{\theta'}\mathcal{L}(\phi', \theta', \mathcal{C})$, and $\epsilon$ is a small scalar. The finite difference is applied to approximate the matrix-vector multiplication term in Eq. 12 for efficient computation.

### 4.3 GRADIENT ANALYSIS

MQ-VAE enhances the gradient guidance for $\mathcal{C}$ by introducing the objective $\mathcal{L}(\phi^*(\mathcal{C}), \theta^*(\mathcal{C}), \mathcal{C})$. While it shares the same functional form as the previous framework (Eq. 6), the arguments $\phi$ and $\theta$ are replaced by the corresponding optimal values $\phi^*(\mathcal{C})$ and $\theta^*(\mathcal{C})$. How the objective makes improvement can be demonstrated by conducting a gradient analysis on the one-step-unrolled surrogate loss with the chain rule. Define $\mathcal{L}'(\phi, \theta, \mathcal{C}) = \mathcal{L}(\phi'(\mathcal{C}), \theta'(\mathcal{C}), \mathcal{C}) = \mathcal{L}(\phi - \xi\nabla_\phi\mathcal{L}(\phi, \theta, \mathcal{C}), \theta - \xi\nabla_\theta\mathcal{L}(\phi, \theta, \mathcal{C}), \mathcal{C})$, we then have

$$\frac{d\mathcal{L}'}{d\mathcal{C}} = \frac{\partial\mathcal{L}'}{\partial\mathcal{C}} + \frac{\partial\phi'}{\partial\mathcal{C}} \times \frac{\partial\mathcal{L}'}{\partial\phi'} + \frac{\partial\theta'}{\partial\mathcal{C}} \times \frac{\partial\mathcal{L}'}{\partial\theta'} \tag{14}$$

The last two terms on the right-hand side, especially $\frac{\partial\phi'}{\partial\mathcal{C}}$ and $\frac{\partial\theta'}{\partial\mathcal{C}}$, referred to as the best-response Jacobian in the literature (Choe et al., 2023), *capture how the encoder-decoder pair reacts to changes of the codebook*. Therefore, the update of $\mathcal{C}$ must consider not only the direct gradient from the loss ($\frac{\partial\mathcal{L}'}{\partial\mathcal{C}}$) for minimizing quantization error but also additional information of indirect gradients about how the encoder and decoder would respond to changes of the codebook ($\frac{\partial\phi'}{\partial\mathcal{C}}$ and $\frac{\partial\theta'}{\partial\mathcal{C}}$), and their performance potential ($\frac{\partial\mathcal{L}'}{\partial\phi'}$ and $\frac{\partial\mathcal{L}'}{\partial\theta'}$). The encoder and decoder choose their best response by conducting gradient descent, which the codebook takes into account. This facilitates finding a globally optimal codebook, thereby enhancing its stability and robustness. See also Figure 2 for the gradient path and hyper-gradient path used in lower level and upper level, respectively. Additionally, we find that in doing so, $\mathcal{C}$ can now receive a gradient from $\mathcal{L}_\text{task}$. For example, the first terms of $\mathcal{L}'$, $\mathcal{L}'_\text{task}$, depends on $\phi'$, which in turn depends on $\mathcal{C}$. The joint effort makes the $\mathcal{L}_\text{task}$ have an influence on $\mathcal{C}$ during backpropagation. In contrast, the original framework updates the $\mathcal{C}$ solely based on the vector quantization objective, which ignores the task loss. We provide a complete derivation of the last two terms in the context of VQ-VAE in Appendix C.

### 4.4 CONNECTION TO EXISTING WORKS

Our method is closely related to two methods proposed in Huh et al. (2023). We show how both methods can be viewed as special cases of MQ-VAE and highlight their insufficiency in achieving the same effects as ours.

**Alternated Optimization**  MQ-VAE can be reduced to an alternated optimization approach by replacing the hyper-gradient at the upper level with a standard gradient. This can be implemented by setting $\xi = 0$ in Algorithm 1, which decouples the two levels in the sense that they are no longer interconnected. By setting $\beta = 0$ (Huh et al., 2023), we obtain the alternated optimization rule:

$$\min_{\mathcal{C}} \mathbb{E}_{(\mathbf{x},\mathbf{y}) \sim \mathcal{D}}[\mathcal{L}_{\text{codebook}}(h_{\mathcal{C}}(F_\phi(\mathbf{x})), F_\phi(\mathbf{x}))] \tag{15}$$

$$\min_{\phi,\theta} \mathbb{E}_{(\mathbf{x},\mathbf{y}) \sim \mathcal{D}}[\mathcal{L}_{\text{task}}(G_\theta(h_{\mathcal{C}}(F_\phi(\mathbf{x}))), \mathbf{y})] \tag{16}$$

This method alternatively updates the codebook and encoder-decoder pair using coordinate descent. Notably, the update steps for both components are standard gradient descent steps with respect to fixed values of each other rather than the hyper-gradient descent described in Algorithm 1. Therefore, $\mathcal{C}$ can not be aware of the subsequent performance of $\phi$ and $\theta$ when updating. Besides, $\mathcal{L}_{\text{task}}$ cannot have a direct impact on $\mathcal{C}$, being the same as VQ-VAE. In contrast, the interconnected nature of bi-level optimization allows side information of the subsequent training of $\phi$ and $\theta$ and makes it possible for $\mathcal{L}_{\text{task}}$ to have a gradient on $\mathcal{C}$.

**Synchronous Update Rule**  Conducting gradient descent on $\mathcal{C}$ using $\mathcal{L}_{\text{codebook}}$ has been shown equivalent to the following exponential moving average (EMA) formula (Van Den Oord et al., 2017):

$$\mathbf{z}_q^{(t+1)} \leftarrow (1 - \xi) \cdot \mathbf{z}_q^{(t)} + \xi \cdot \mathbf{z}_e^{(t)} \tag{17}$$

where $\xi$ is the learning rate. The synchronous update rule (Huh et al., 2023) modifies the EMA update rule by using $\mathbf{z}_e^{(t+1)}$, the feature embedding after the current EMA update step instead:

$$\mathbf{z}_q^{(t+1)} \leftarrow (1 - \xi) \cdot \mathbf{z}_q^{(t)} + \xi \cdot \mathbf{z}_e^{(t+1)} \tag{18}$$

which can be explicitly expressed as

$$\mathbf{z}_q^{(t+1)} \leftarrow (1 - \xi) \cdot \mathbf{z}_q^{(t)} + \xi \cdot \mathbf{z}_e^{(t)} + \xi^2 \cdot \frac{\partial \mathcal{L}_{\text{task}}}{\partial \mathbf{z}_q} \tag{19}$$

The additional term $\xi^2 \cdot \frac{\partial \mathcal{L}_{\text{task}}}{\partial \mathbf{z}_q}$ introduces a gradient from $\mathcal{L}_{\text{task}}$ to $\mathbf{z}_q$, a subset of $\mathcal{C}$.

In fact, our direct hyper-gradient term $\frac{\partial \mathcal{L}'}{\partial \mathcal{C}}$ can achieve the same effect as the synchronous update rule. To see this, a straightforward computation shows

$$\frac{\partial \mathcal{L}'(\phi, \theta, \mathcal{C})}{\partial \mathcal{C}} = \frac{\partial \mathcal{L}(\phi', \theta', \mathcal{C})}{\partial \mathcal{C}} = \frac{\partial \mathcal{L}_{\text{codebook}}(\phi', \theta', \mathcal{C})}{\partial \mathcal{C}} \tag{20}$$

Here, we are computing the partial derivative with respect to $\mathcal{C}$, and only the codebook loss $\mathcal{L}_{\text{codebook}}$ has a gradient. We can draw the observation that solely applying the direct hyper-gradient is equivalent to updating the codebook with embedding generated by the updated encoder. That is applying the EMA update rule with $\mathbf{z}_e^{(t+1)}$, the embedding after one step update, being the same as the synchronous update rule. Importantly, the direct hyper-gradient part is insufficient for the codebook to be aware of the subsequent performance of the encoder and decoder, as illustrated in Section 4.3.

## 5  EXPERIMENT

### 5.1  EXPERIMENTAL SETUP

We compare MQ-VAE with the traditional vector quantization framework used in VQ-VAE and other variants, including the least-recently-used (LRU) replacement policy (denoted as '+replace', Łańcucki et al. (2020); Zeghidour et al. (2021); Dhariwal et al. (2020)) and grouped latent variables (denoted as '+group', Kaiser et al. (2018); Guo et al. (2024); Yu et al. (2021)) . For comparison, we combine the variants MQ-VAE and VQ-VAE, respectively. We also include a differentiable quantization baseline Gumbel-VQ (Karpathy, 2021) for direct comparison, whose codebook can be optimized using standard backpropagation. All models were initialized using the K-means clustering algorithm. Fashion-MNIST (Xiao et al., 2017) and CIFAR100 (Krizhevsky et al., 2009) are used for evaluation. Appendix B provides additional results on MNIST and SVHN datasets. Our implementation is mainly based on the VQTorch library (Huh, 2022) for the compared baselines and the Betty library (Choe et al., 2023) for an efficient bi-level optimization algorithm. A single NVIDIA 3090 GPU is used for all experiments.

Table 1: **Generative modeling:** Comparison on image reconstruction tasks. FID, IS and LPIPS all denote the quality of reconstructed image.

| Dataset | Method | FID ↓ | IS ↑ | LPIPS $(10^{-1})$ ↓ | MSE $(10^{-3})$ ↓ | Perplexity ↑ | $\Delta_{\text{gap}}(10^{-5})$ ↓ |
|---------|--------|-------|------|---------------------|-------------------|--------------|----------------------------------|
| Fashion MNIST | VQ-VAE | 17.3 | 3.95 | 0.59 | 7.26 | 50.8 | 17.68 |
| | Gumbel-VQVAE | 77.2 | 3.47 | 1.11 | 13.07 | **118.4** | 47.59 |
| | MQ-VAE | **7.9** | **4.14** | **0.33** | **4.28** | 80.5 | **8.63** |
| | VQ-VAE + replace | 9.8 | 4.12 | 0.38 | 4.25 | 303.4 | 9.68 |
| | MQ-VAE + replace | **6.8** | **4.18** | **0.28** | **2.90** | **356.8** | **4.71** |
| | VQ-VAE + group | 4.1 | 4.33 | 0.15 | 1.93 | **121.3** | 5.97 |
| | MQ-VAE + group | **2.4** | **4.35** | **0.06** | **0.73** | 88.9 | **1.30** |
| CIFAR100 | VQ-VAE | 81.8 | 4.99 | 1.88 | 7.73 | 54.4 | 8.71 |
| | Gumbel-VQVAE | 144.3 | 3.16 | 2.41 | 20.42 | **121.4** | 36.60 |
| | MQ-VAE | **53.5** | **6.4** | **1.25** | **5.03** | 63.4 | **5.83** |
| | VQ-VAE + replace | 49 | 6.9 | 1.09 | 4.19 | 699.9 | 4.44 |
| | MQ-VAE + replace | **41.8** | **7.43** | **0.9** | **3.19** | **819.5** | **2.12** |
| | VQ-VAE + group | 29.1 | 8.65 | 0.61 | 2.86 | 103.7 | 4.04 |
| | MQ-VAE + group | **9.2** | **11.48** | **0.14** | **0.77** | **222.2** | **0.48** |

## 5.2 GENERATIVE MODELING

In a generative modeling task, we first train the codebook through self-supervised reconstruction, then freeze the pre-trained codebook and use it for downstream tasks, such as image generation, following Van Den Oord et al. (2017). For performance metrics, We use the Inception Score (IS, Salimans et al. (2016)), Fréchet Inception Distance (FID, Heusel et al. (2017)), LPIPS perceptual loss (Zhang et al., 2018), and mean squared error (MSE, as task loss for reconstruction task). Additionally, we report perplexity and gradient estimation gap (Huh et al., 2023). The perplexity is defined as $2^{H(p)}$, where $H(p)$ is the entropy of the codebook's probability distribution. A higher perplexity implies a more uniform assignment of codes, indicating a higher code utilization rate. The gradient estimation gap is defined as

$$\Delta_{\text{gap}} = \left\| \frac{\partial \mathcal{L}_{\text{task}}(G_\theta(\mathbf{z}_e))}{\partial F_\phi(\mathbf{x})} - \frac{\partial \mathcal{L}_{\text{task}}(G_\theta(\mathbf{z}_q))}{\partial F_\phi(\mathbf{x})} \right\|,$$

which measures the difference between the gradients of the non-quantized model and the quantized model. A zero gap implies that the gradient descent using STE is guaranteed to minimize the loss; thus, the lower the gap, the better.

The results presented in Table 1 demonstrate that MQ-VAE achieves the best performance across all evaluation metrics. Consistent with Huh et al. (2023) and Zheng & Vedaldi (2023), our results reveal that higher perplexity does not necessarily imply better performance. Instead of explicitly encouraging a high codebook utilization rate, our codebook is meta-learned based on the subsequent performance of the encoder and decoder. By maintaining a balanced codebook utilization rate that avoids both under-utilization and redundancy, our method significantly outperforms baseline methods. The improvement in the gradient estimation gap demonstrates that MQ-VAE has a stronger ability to enhance the subsequent gradient estimation of the encoder and decoder, resulting in more stable training. Additionally, the improvement in MSE indicates that introducing gradients from $\mathcal{L}_{\text{task}}$ to $\mathcal{C}$ is crucial for achieving low task loss, which in turn improves performance metrics.

In Figure 3, we perform image generation using PixelCNN (Van Den Oord et al., 2016) as the prior alongside the pre-trained codebook in the reconstruction task. We plot the curves of FID during training on both Fashion-MNIST and SVHN datasets. The results indicate that, by achieving lower task loss and improved performance in the reconstruction task, MQ-VAE learns a significantly more powerful codebook for downstream tasks. Specifically, the meta-learned codebook yields better performance than VQ-VAE across both datasets. This suggests that our framework can produce a more generalizable and versatile codebook as a discrete representation by enabling awareness of the performance of other model components such the encoder and decoder.

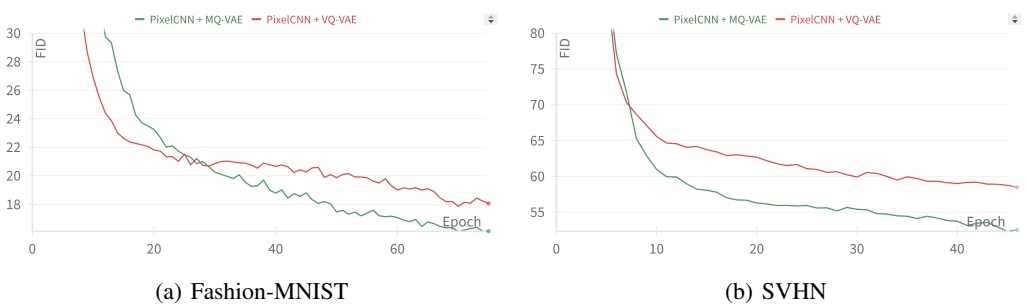

(a) Fashion-MNIST

(b) SVHN

Figure 3: PixelCNN training curves on Fashion-MNIST and SVHN dataset, with FID reported ( The lower, the better).

Table 2: **Classification:** The effect of how our methods affect the final performance on classification.

| Dataset | Method | Accuracy ↑ | F1 Score ↑ | CE Loss ↓ | Perplexity ↑ | $\Delta_{\text{gap}}$ ↓ |
|---|---|---|---|---|---|---|
| Fashion MNIST | VQ-VAE | 89.1 | 89 | 0.95 | 112.1 | 1.38 |
| | Gumbel-VQVAE | 88.7 | 88.7 | 0.97 | **718.7** | **0.98** |
| | MQ-VAE | **89.6** | **89.5** | **0.84** | 82.9 | 1.14 |
| | VQ-VAE + replace | 89.2 | 89.2 | 1.00 | 421.1 | 1.51 |
| | MQ-VAE + replace | **89.9** | **89.8** | **0.91** | **443.9** | **1.29** |
| | VQ-VAE + group | 89.2 | 89.2 | 0.86 | 15.8 | 4.25 |
| | MQ-VAE + group | **89.4** | **89.4** | **0.9** | **29.5** | **4.08** |
| CIFAR100 | VQ-VAE | 24.5 | 25.9 | **5.36** | 203.8 | 3.76 |
| | Gumbel-VQVAE | 26.8 | 26.6 | 9.50 | **938.7** | 2.95 |
| | MQ-VAE | **28.2** | **29.2** | 6.58 | 97.3 | **1.93** |
| | VQ-VAE + replace | 28.1 | 29.0 | 5.18 | 420.9 | 4.08 |
| | MQ-VAE + replace | **33.8** | **33.9** | **4.69** | **426.2** | **1.35** |
| | VQ-VAE + group | 29.2 | 29.4 | 5.58 | **155.3** | 5.41 |
| | MQ-VAE + group | **30.3** | **30.6** | **5.46** | 73.2 | **4.64** |

## 5.3 CLASSIFICATION TASK

We also apply our method to the classification task following Huh et al. (2023), using top-1 accuracy and top-1 F1 score as performance metrics. Cross-entropy (CE) loss is used to calculate task loss. ResNet18 (He et al., 2016) is used as the backbone model and is quantized after the second macroblock, which is roughly the halfway point in ResNet18. Other settings remain the same as in previous work.

The results presented in Table 2 show that similar to the previous section, MQ-VAE achieves the best or comparable performance across all evaluation metrics except perplexity. This means our method enhances performance without relying solely on high perplexity. MQ-VAE significantly reduces the gradient estimation gap and benefits the subsequent optimization of the encoder and decoder. Direct gradient guidance from $\mathcal{L}_{\text{task}}$ to $\mathcal{C}$ results in lower CE loss and, in turn, better accuracy and F1 score. This demonstrates the superiority of MQ-VAE in training quantization codebooks for classification tasks.

## 5.4 ABLATION STUDIES

We conduct ablation studies using the two methods mentioned in Section 4.4: alternated optimization (+alt) and the synchronous update rule (+sync). We also include their direct combination (+alt+sync) for reference. Table 3 presents the results of the generative modeling task with the same experimental setup as in Section 5.2. The results show that MQ-VAE outperforms ablation

Table 3: **Ablation studies:** Comparison between MQ-VAE and ablation baselines on image reconstruction task. FID, IS, and LPIPS all denote the quality of the reconstructed image.

| Dataset | Method | FID ↓ | IS ↑ | LPIPS $(10^{-1})$ ↓ | MSE $(10^{-3})$ ↓ | Perplexity ↑ | $\Delta_{\mathrm{gap}}(10^{-5})$ ↓ |
|---|---|---|---|---|---|---|---|
| Fashion MNIST | VQ-VAE | 17.3 | 3.95 | 0.59 | 7.26 | 50.8 | 17.68 |
| | VQ-VAE + alt | 14.4 | 3.97 | 0.53 | 5.96 | 56.7 | 13.28 |
| | VQ-VAE + sync | 11.4 | **4.14** | 0.46 | 7.00 | 55 | 14.5 |
| | VQ-VAE + alt + sync | 9.8 | 4.07 | 0.42 | 4.73 | **95.3** | **1.41** |
| | MQ-VAE | **7.9** | **4.14** | **0.33** | **4.28** | 80.5 | 8.63 |
| CIFAR100 | VQ-VAE | 81.8 | 4.99 | 1.88 | 7.73 | 54.4 | 8.71 |
| | VQ-VAE + alt | 55.6 | 5.91 | 1.32 | 6.29 | 271.7 | **0.42** |
| | VQ-VAE + sync | 69.1 | 5.55 | 1.56 | 7.68 | 59.3 | 10.63 |
| | VQ-VAE + alt + sync | 55.3 | 5.84 | 1.32 | 6.26 | **271.8** | **0.42** |
| | MQ-VAE | **53.5** | **6.4** | **1.25** | **5.02** | 63.4 | 5.83 |

baselines in terms of all generation quality measures. We find that although alternated optimization can achieve higher codebook perplexity and lower gradient estimation gap, it does not necessarily improve generation performance. We conjecture this is because alternated optimization solely concentrates on reducing current quantization error without explicitly considering task performance. On the other hand, the synchronous update rule can improve performance by introducing a gradient from task loss but does not account for the subsequent training of the encoder and decoder. The superiority of MQ-VAE over the two baselines and their combination demonstrates the effectiveness of our bi-level optimization framework. The richer gradient flow significantly enhances codebook training by balancing the objectives of enhancing task performance and improving encoder and decoder training.

## 5.5 COMPUTATION COSTS

MQ-VAE shares the same network architecture as VQ-VAE but requires additional forward and backward passes at the lower level for hyper-gradient calculation, as shown in Eq. 13. This results in higher computational costs compared to VQ-VAE. Table 4 presents an empirical comparison of the average training costs between MQ-VAE and VQ-VAE when trained for the same number of iterations. Importantly, we found that MQ-VAE converges significantly faster than the baseline, as indicated by the green dashed lines in Figure 4. This mitigates the disadvantage associated with speed, allowing MQ-VAE to achieve comparable performance in the same or less wall-clock time. Therefore, MQ-VAE is practical due to its superior performance and comparable speed.

Table 4: Average training time comparison on the reconstruction task, with the cost of VQ-VAE normalized to 1 for reference.

| Method | VQ-VAE | MQ-VAE |
|---|---|---|
| Cost | ×1 | ×3.95 |

## 6 CONCLUSION

We propose MQ-VAE, a novel bi-level optimization-based vector-quantization framework inspired by meta-learning. Following VQ-VAE, our method trains the codebook and encoder-decoder pair within a cohesive bi-level optimization problem. Without fully relying on the heuristic assumption about codebook utilization rate, our approach ensures that the codebook's objective minimizes not only the quantization error but also enhances subsequent training of the encoder and decoder. Additionally, compared to the vanilla vector

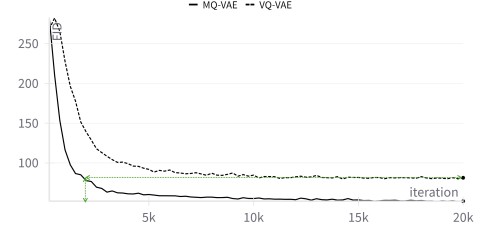

Figure 4: MQ-VAE reaches the same FID score using much less time when compared with VQ-VAE.

quantization objective, MQ-VAE facilitates a gradient flow from the task loss to the codebook, thereby improving overall performance. Empirical studies across various computer vision tasks demonstrate that MQ-VAE outperforms existing methods and ablation baselines, underscoring its effectiveness.

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

## A    MORE RELATED WORK ON VECTOR-QUANTIZED NETWORKS

The vector-quantization layer in deep learning was first introduced in generative models as the vector-quantized variational auto-encoder (VQ-VAE), which maps continuous embedding to discrete vectors using a learned codebook. Since the quantization operation is not differentiable, straight-through estimation (Bengio et al., 2013) is applied to allow gradients to flow back through the VQ layer by ignoring it during the backward pass. The codebook is learned using the vector-quantization objective, where an $l_2$ error is applied to adjust the selected codebook entries toward the corresponding embedding. However, several issues arise from this design: 1) index collapse, where only a small fraction of codes are utilized during training; 2) a gradient estimation gap incurred by quantization error at the VQ layer, leading to biased and unstable gradient descent; and 3) the gradient of the task loss (e.g., reconstruction loss in generative modeling) does not propagate to the codebook.

Numerous efforts have been made to address these problems. Van Den Oord et al. (2017) and Razavi et al. (2019) propose an exponential moving averages (EMA) approach for codebook training. Similar to K-Means, the EMA gradually moves the selected codewords toward the centroids of encoder outputs. Łańcucki et al. (2020), Zeghidour et al. (2021), and Dhariwal et al. (2020) introduce a reset mechanism for codewords that have not been selected for an extended period, updating them with embedding from the current batch to enhance codebook utilization. Roy et al. (2018) and Takida et al. (2022) present probabilistic reformulations of VQ-VAE, wherein the quantization step is made stochastic rather than relying on the nearest neighbor approach, allowing non-neighboring codewords to be selected and improving utilization. Lee et al. (2022) introduces a residual-quantization method that reduces quantization error by recursively applying quantization operation on the quantization error to better approximate feature maps. Other approaches, such as Kaiser et al. (2018), Guo et al. (2024), and Yu et al. (2021), involve breaking up (or projecting) embedding and codewords into smaller slices, applying the same quantization process, and then recovering the quantized features by concatenating the slices. Huh et al. (2023) proposes an affine reparameterization of codewords, allowing gradients to flow through unselected code-vectors via affine parameters, as affined codewords (a weighted sum of all codewords) are used for nearest neighbor searching. Yu et al. (2021) applies $l_2$ normalization to both embedding and codebook latent variables, effectively mapping all latent variables onto a sphere and replacing Euclidean distance with cosine similarity, thus unifying the scale of latent variables and enhancing training stability. Sønderby et al. (2017) and Karpathy (2021) suggest a continuous approximation of vector quantization, making the bottleneck differentiable and allowing standard backpropagation training. Gumbel-VQ, introduced by Karpathy (2021) and used as a comparison method in Huh et al. (2023), minimizes the ELBO and, unlike traditional VQ methods, predicts a distribution over the code without explicit distance comparisons. The Gumbel-softmax (Jang et al., 2016) trick is then employed to sample from this distribution.

## B    MORE RESULTS ON GENERATIVE MODELING AND ABLATION STUDIES

In addition to the results of the generative modeling task in Section 5.2 and the ablation studies in Section 5.4, we further apply our method to the SVHN (Netzer et al., 2011) and CIFAR10 (Krizhevsky et al., 2009) datasets. The results presented in Table 5 and Table 6 show consistent performance improvements to previous sections, showing the effectiveness of our method. The reasons for these enhancements are similar to the explanations provided earlier.

Table 5: **Generative modeling reconstruction:** Comparison between various methods on image reconstruction tasks. FID, IS and LPIPS all denote the quality of reconstructed image.

| Dataset | Method | FID ↓ | IS ↑ | LPIPS $(10^{-1})$ ↓ | MSE $(10^{-3})$ ↓ | Perplexity ↑ | $\Delta_{\text{gap}}(10^{-5})$ ↓ |
|---|---|---|---|---|---|---|---|
| SVHN | VQ-VAE | 100.2 | 2.19 | 0.97 | 3.24 | 38.3 | 7.19 |
| | Gumbel-VQVAE | 71.8 | 2.49 | 1.08 | 4.36 | **237.6** | 12.90 |
| | MQ-VAE | **53.1** | **2.5** | **0.51** | **1.49** | 67.2 | **2.37** |
| | VQ-VAE + replace | 54 | 2.54 | 0.42 | 1.03 | 430.5 | 2.08 |
| | MQ-VAE + replace | **44.7** | **2.7** | **0.32** | **0.71** | **522.6** | **0.88** |
| | VQ-VAE + group | 45.3 | 2.6 | 0.4 | 1.15 | 71.3 | 3.79 |
| | MQ-VAE + group | **12** | **2.99** | **0.14** | **0.34** | **79.2** | **1.20** |
| CIFAR10 | VQ-VAE | 72 | 5.04 | 1.77 | 7.58 | 56.2 | 9.20 |
| | Gumbel-VQVAE | 203.6 | 2.71 | 3.44 | 21.89 | **115.8** | 33.78 |
| | MQ-VAE | **58.9** | **5.43** | **1.44** | **5.82** | 63.4 | **7.01** |
| | VQ-VAE + replace | 45.9 | **6.41** | 1.01 | 3.94 | 800.4 | 4.82 |
| | MQ-VAE + replace | **42.9** | 6.37 | **0.98** | **3.75** | **846.6** | **4.66** |
| | VQ-VAE + group | 26.9 | 7.43 | 0.57 | 2.74 | 110.8 | 4.13 |
| | MQ-VAE + group | **8.1** | **9.41** | **0.13** | **0.65** | **252.3** | **0.44** |

Table 6: **Ablation studies:** Comparison between MQ-VAE and ablation baselines on image reconstruction task. FID, IS, and LPIPS all denote the quality of the reconstructed image.

| Dataset | Method | FID ↓ | IS ↑ | LPIPS $(10^{-1})$ ↓ | MSE $(10^{-3})$ ↓ | Perplexity ↑ | $\Delta_{\text{gap}}(10^{-5})$ ↓ |
|---|---|---|---|---|---|---|---|
| SVHN | VQ-VAE | 100.2 | 2.19 | 0.97 | 3.24 | 38.3 | 7.19 |
| | VQ-VAE + alt | 90.7 | 2.28 | 0.75 | 2.38 | 59.1 | 4.49 |
| | VQ-VAE + sync | **44.6** | 2.43 | 0.74 | 2.66 | 50.7 | 4.42 |
| | VQ-VAE + alt + sync | 85.3 | **2.82** | 0.58 | 1.67 | 63.2 | **0.77** |
| | MQ-VAE | 53.1 | 2.5 | **0.51** | **1.48** | **67.2** | 2.37 |
| CIFAR10 | VQ-VAE | 72 | 5.04 | 1.77 | 7.57 | 56.2 | 9.20 |
| | VQ-VAE + alt | 69.9 | 5.06 | 1.44 | 6.23 | 60.4 | **4.56** |
| | VQ-VAE + sync | 65.9 | 5.33 | 1.6 | 7.70 | 57.2 | 10.6 |
| | VQ-VAE + alt + sync | 65.5 | 5.38 | **1.40** | 6.18 | 62.5 | 7.62 |
| | MQ-VAE | **58.9** | **5.43** | 1.44 | **5.81** | **63.4** | 7.01 |

## C    DERIVATION OF GRADIENT FOR CODEBOOK UPDATING

We provide a complete derivation of the gradient updating rule in Eq. 14 as follows. Apply the backpropagation chain of VQ-VAE, we get

$$\frac{\partial \mathcal{L}_{\text{task}}}{\partial \phi} = \frac{\partial \mathcal{L}_{\text{task}}}{\partial \mathbf{y}} \cdot \frac{\partial \mathbf{y}}{\partial \mathbf{z}_q} \cdot \frac{\partial \mathbf{z}_e}{\partial \phi} \tag{21}$$

$$\frac{\partial \mathcal{L}_{\text{task}}}{\partial \theta} = \frac{\partial \mathcal{L}_{\text{task}}}{\partial \mathbf{y}} \cdot \frac{\partial \mathbf{y}}{\partial \theta} \tag{22}$$

$$\frac{\partial \mathcal{L}_{\text{commit}}}{\partial \phi} = \frac{\partial \mathcal{L}_{\text{commit}}}{\partial \mathbf{z}_e} \cdot \frac{\partial \mathbf{z}_e}{\partial \phi} \tag{23}$$

Recall $\mathcal{L} = \mathcal{L}_{\text{task}} + \mathcal{L}_{\text{codebook}} + \mathcal{L}_{\text{commit}}$, we get

$$\frac{\partial \mathcal{L}}{\partial \phi} = \left( \frac{\partial \mathcal{L}_{\text{task}}}{\partial \mathbf{y}} \cdot \frac{\partial \mathbf{y}}{\partial \mathbf{z}_q} + \frac{\partial \mathcal{L}_{\text{commit}}}{\partial \mathbf{z}_e} \right) \cdot \frac{\partial \mathbf{z}_e}{\partial \phi} \tag{24}$$

$$\frac{\partial \mathcal{L}}{\partial \theta} = \frac{\partial \mathcal{L}_{\text{task}}}{\partial \mathbf{y}} \cdot \frac{\partial \mathbf{y}}{\partial \theta} \tag{25}$$

$$\frac{\partial \mathcal{L}\prime}{\partial \phi'} = \left( \frac{\partial \mathcal{L}'_{\text{task}}}{\partial \mathbf{y}'} \cdot \frac{\partial \mathbf{y}'}{\partial \mathbf{z}'_q} + \frac{\partial \mathcal{L}'_{\text{commit}}}{\partial \mathbf{z}'_e} \right) \cdot \frac{\partial \mathbf{z}'_e}{\partial \phi'} \tag{26}$$

$$\frac{\partial \mathcal{L}'}{\partial \theta'} = \frac{\partial \mathcal{L}'_{\text{task}}}{\partial \mathbf{y}'} \cdot \frac{\partial \mathbf{y}'}{\partial \theta'} \tag{27}$$

Plugin in $\phi' = \phi - \xi \nabla_\phi \mathcal{L}(\phi, \theta, \mathcal{C})$ and $\theta' = \theta - \xi \nabla_\theta \mathcal{L}(\phi, \theta, \mathcal{C})$,

$$\frac{\partial \phi'}{\partial \mathcal{C}} = -\xi \frac{\partial^2 \mathcal{L}}{\partial \mathcal{C} \partial \phi} = -\xi \frac{\partial \mathbf{z}_e}{\partial \phi} \cdot \frac{\partial \left( \frac{\partial \mathcal{L}_{\text{task}}}{\partial \mathbf{y}} \cdot \frac{\partial \mathbf{y}}{\partial \mathbf{z}_q} + \frac{\partial \mathcal{L}_{\text{commit}}}{\partial \mathbf{z}_e} \right)}{\partial \mathcal{C}} \tag{28}$$

$$\frac{\partial \theta'}{\partial \mathcal{C}} = -\xi \frac{\partial^2 \mathcal{L}}{\partial \mathcal{C} \partial \phi} = -\xi \frac{\partial \left( \frac{\partial \mathcal{L}_{\text{task}}}{\partial \mathbf{y}} \cdot \frac{\partial \mathbf{y}}{\partial \theta} \right)}{\partial \mathcal{C}} \tag{29}$$

## D    A GAME THEORY PERSPECTIVE OF MQ-VAE

Our method can also be interpreted within the framework of Stackelberg games (Von Stackelberg, 2010; Rajeswaran et al., 2020). Stackelberg games are asymmetric games that impose a specific order of play and generalize min-max games. Consider two players, $A$ and $B$, with parameters $\boldsymbol{\theta}_A$ and $\boldsymbol{\theta}_B$. Each player aims to minimize their losses $\mathcal{L}_A(\boldsymbol{\theta}_A, \boldsymbol{\theta}_B)$ and $\mathcal{L}_B(\boldsymbol{\theta}_A, \boldsymbol{\theta}_B)$. With player $A$ as the leader, the Stackelberg game corresponds to the following nested optimization:

$$\min_{\boldsymbol{\theta}_A} \mathcal{L}_A(\boldsymbol{\theta}_A, \boldsymbol{\theta}_B^*(\boldsymbol{\theta}_A))$$
$$s.t. \quad \boldsymbol{\theta}_B^*(\boldsymbol{\theta}_A) = \arg \min_{\boldsymbol{\theta}_B} \mathcal{L}_B(\boldsymbol{\theta}_A, \boldsymbol{\theta}_B) \tag{30}$$

Our problem structure aligns with Eq. 30 by viewing the codebook as the leader and the encoder-decoder pair as the follower. The follower's parameters depend implicitly on the leader's parameters, which the leader can exploit when updating its parameters. In this way, the leader will not only minimize its loss but also the influence on the later updating of the follower. A detailed illustration of this point in our context has been explained in Section 4.3.

## E    BI-LEVEL OPTIMIZATION DESIGN CHOICES

Several design choices exist for applying a hyper-gradient in training VQ-VAE, including applying it only to the codebook (MQ-VAE, the method presented in the main paper), only to the encoder-decoder pair, or to both sides. Our design choice is motivated by several factors. First, in existing literature, the upper layer typically contains far fewer parameters than the lower layer. Relevant examples include neural architecture search (Liu et al., 2018; Zhang et al., 2021) and hyperparameter

Table 7: We compare three design choices for applying hyper-gradients. MQ-VAE[†] applies hyper-gradients only to the encoder-decoder pair, while MQ-VAE[‡] applies hyper-gradients to both the encoder-decoder pair and the codebook. The best results are indicated in **bold**, and the second-best results are shown in *italic*.

| Dataset | Method | FID $\downarrow$ | IS $\uparrow$ | LPIPS $(10^{-1})\downarrow$ | MSE $(10^{-3})\downarrow$ |
|---|---|---|---|---|---|
| CIFAR100 | VQ-VAE | 81.8 | 4.99 | 1.88 | 7.73 |
| | MQ-VAE | *53.5* | **6.40** | *1.25* | *5.02* |
| | MQ-VAE[†] | 56.1 | 5.93 | 1.44 | 5.48 |
| | MQ-VAE[‡] | **52.8** | **6.40** | **1.23** | **4.58** |

optimization (HPO) (Lorraine et al., 2020; Franceschi et al., 2017). Additionally, during VQ-VAE training, the encoder-decoder pair can often be optimized more smoothly, due to the sparse gradient characteristics of codebook training. By enabling the codebook to "see into the future," we provide it with additional information, facilitating a more balanced interaction between the two components and allowing the codebook to also account for the task loss. Besides, when hyper-gradient is applied to the codebook, it shows a strong connection to existing work after a close inspection, which further justifies this design choice. We did not find an obvious connection when the hyper-gradient is applied to the other side.

In principle, hyper-gradient descent could be applied to both the encoder-decoder pair and the codebook by using a surrogate loss function for each. This approach resembles policy prediction (Zhang & Lesser, 2010) in multi-agent learning, which is known to achieve convergence in games where gradient descent (ascent) fails. Applying a similar idea to our problem allows both the autoencoder and the codebook to predict each other's future states, potentially enabling more robust training. However, this comes with significantly higher computational costs, and we did not use this strategy in the main paper for simplicity.

We conduct an empirical comparison between the three choices and present the results in Table 7. We evaluate the reconstruction performance on the CIFAR100 dataset. We can draw the observation that using hyper-gradient descent for the codebook does not result in performance improvements comparable to MQ-VAE. Besides, while using hyper-gradient descent for both sides achieves slightly improved performance, it is less computationally efficient, which is roughly twice as expensive as the original MQ-VAE in principle.

## F    IMPORTANCE OF RETAINING COMPUTATION GRAPH OF LOWER LEVEL TRAINING

The concept of differentiating through the lower-level training process is crucial in MQ-VAE. If the computation graph of lower-level training is disconnected such that $\phi^*$ and $\theta^*$ are no longer functions of $\mathcal{C}$, the desired effect cannot be achieved. Specifically, our method cannot be realized through a $k$-step look-ahead procedure, which involves training the encoder-decoder for $k$ steps before updating the codebook without retaining the computation graph. After updating the codebook, the $k$ updates to the encoder-decoder are undone, and we perform one new update step instead. However, the key difference lies in backpropagating the learning signal through the unrolling in Eq. 10. This backpropagation can be controlled by introducing stop gradient calls into the computation graph between unrolling steps. When a stop gradient is applied, $\phi^*$ and $\theta^*$ are treated independently of $\mathcal{C}$, resulting in an update signal that corresponds only to the first term in Eq. 14, which is insufficient for optimizing the codebook effectively.

## G    SCALING UP TO LARGER EXPERIMENTAL SETTINGS

We evaluate the scalability of our method using the modern architecture VQ-GAN (Esser et al., 2021). In this context, our meta-learning-based VQ-GAN is referred to as MQ-GAN. The network architecture directly follows the original VQ-GAN paper and their GitHub repository[1]. We also

---

[1]https://github.com/CompVis/taming-transformers

Table 8: **Generative modeling reconstruction:** Comparison between various methods on image reconstruction tasks.

| Dataset | Method | FID ↓ | IS ↑ | LPIPS $(10^{-1})$ ↓ | Perceptual loss ↓ | Perplexity ↑ |
|---------|--------|-------|------|---------------------|-------------------|--------------|
| CelebA-HQ | VQ-GAN | 25.1 | 2.90 | 1.34 | 0.242 | 79.5 |
| | MQ-GAN | **23.2** | **2.97** | **1.28** | **0.218** | **79.7** |
| Imagenet | VQ-GAN | 34.5 | 1.83 | 1.26 | 0.163 | 77.4 |
| | MQ-GAN | **30.7** | **1.92** | **1.15** | **0.149** | **80.8** |

use larger datasets, including CelebA-HQ (Karras, 2017) and ImageNet (Deng et al., 2009), both with a resolution of 256 × 256. Multiple GPU devices (4 NVIDIA-A10 GPUs for the CelebA-HQ dataset and 2 NVIDIA-A10 GPUs for the ImageNet dataset) are employed for distributed training, which better aligns with practical scenarios. Due to limited computational resources, we train both methods for only 10k and 5k iterations, respectively. Although the numerical results may not reach those reported in the original VQ-GAN paper, the comparisons remain fair.

The results in Table 8 demonstrate that our method achieves superior performance in larger-scale experimental settings. Additionally, we provide samples reconstructed by VQ-GAN and MQ-GAN in Figure 5. The results qualitatively reveal that our method can reconstruct images with higher quality.

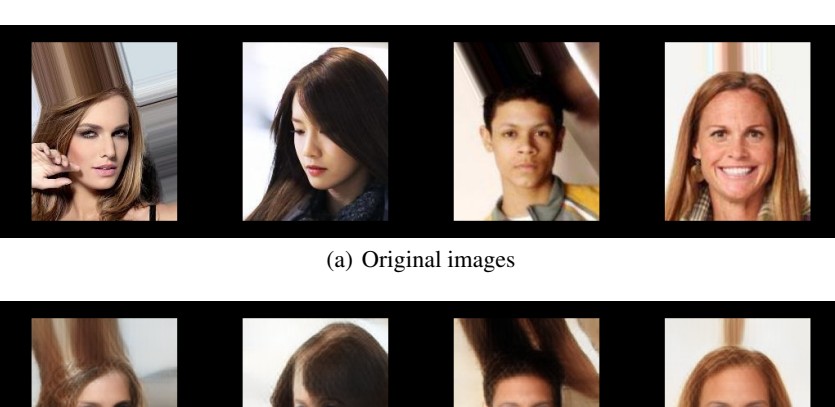

(a) Original images

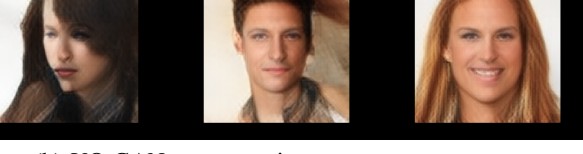

(b) VQ-GAN reconstruction

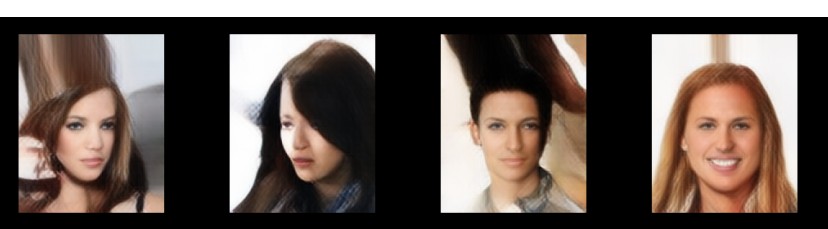

(c) MQ-GAN reconstruction

Figure 5: Reconstruction samples on Imagenet dataset

## H EMPIRICAL GRADIENT ANALYSIS

We explore the importance of hyper-gradients by comparing the dynamics of indirect and direct hyper-gradients. Figure 6 shows the norms for both parts and their cosine similarity. From the results, we observe that in terms of the $L^2$ norm, the magnitudes of direct and indirect gradients are

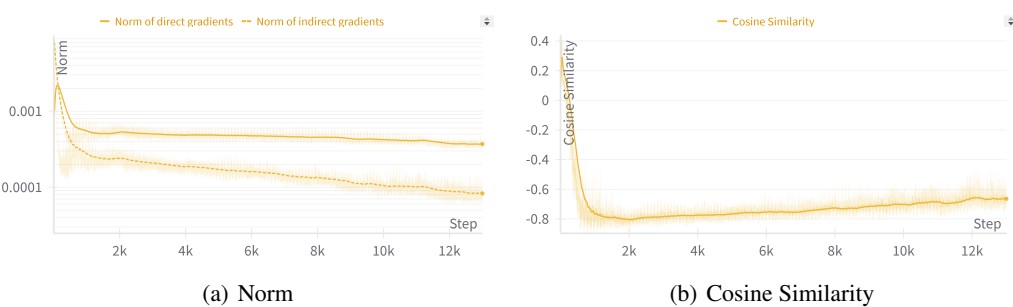

(a) Norm                     (b) Cosine Similarity

Figure 6: The gradient dynamics when MQ-VAE evaluate on the CIFAR100 dataset. A similar pattern can be observed for other datasets.

comparable. Specifically, the norm of direct gradients is only around four times larger than that of the indirect gradients, meaning the effects of indirect gradients should not be ignored. Moreover, the cosine similarity of the two gradients reveals that their effects are not fully correlated. Thus, the gradient guidance of the codebook can be significantly enriched by the indirect hyper-gradient. Additionally, we can observe that in the middle stage, the indirect gradients tend to have a contrary effect, which may act as implicit regularization and make the training more robust. In conclusion, both parts of the hyper-gradients make a significant contribution to codebook training.

## I  IMPLEMENTATION DETAILS

We use $1024$ codewords for all our experiments, following the typical range of $1024 \sim 4096$ codewords in prior works (Yan et al., 2021; Huh et al., 2023). Each codeword has $512$ dimensions. The trade-off parameter $\beta$ was set to $0.1$ for calculating the VQ loss. The data preprocessing procedures are simple and straightforward: we convert the image into a tensor with a value range of [0, 1]. For the ImageNet dataset, where the images are not square, we crop the central $256 \times 256$ pixels. All image quality metrics are computed using the torch-metrics library[2]. For the reconstruction task, the backbone architecture directly follows the original VQ-VAE paper (Van Den Oord et al., 2017). For the classification task, we use the standard ResNet18 architecture from the PyTorch library, with the quantization layer inserted after the second macroblock.

For training, we used the AdamW optimizer (Loshchilov, 2017) with a linear warmup with cosine annealing learning rate scheduler and a warmup ratio of $0.1$ for all experiments. A batch size of $512$ was used consistently across all experiments. The training configurations for other baselines were kept the same as those for our method. The learning rate for the lower layer (i.e., the encoder-decoder pair) was set to $3 \times 10^{-4}$, and the learning rate for the upper layer (i.e., the codebook) was set to $6 \times 10^{-2}$. A weight decay of $1 \times 10^{-4}$ was applied to both layers.

---

[2]https://lightning.ai/docs/torchmetrics/stable/