# OpenReview forum: "MQ-VAE: Training Vector-Quantized Networks via Meta Learning"
_ICLR.cc/2025/Conference — Submitted to ICLR 2025_

### Official Review · Reviewer_rpsn · 2024-10-18

**Soundness:** 3
**Presentation:** 3
**Contribution:** 2
**Rating:** 3
**Confidence:** 4

**Summary:**

**Background**: Gradient-based meta-learning (GBML) aims to updated a models' parameters $\theta$ such that, after only several gradient updates with respect to a class in the support set, the loss on this task is minimized. Intuitively, it tries to find a good set of model parameters that can be quickly adapted (after a couple finetuning steps) to a downstream distribution.

**This work.** This paper proposes to adapt gradient-based meta-learning (GBML) from the typical few-shot learning setting into the training of a VQ-VAE.

Specifically, the encoder parameters $\phi$ and decoder parameters $\theta$ are trained with a fixed codebook for several steps, the loss is computed with the updated encoder/decoder parameters, and the codebook is updated with respect to this loss. As in GBML, the encoder/decoder parameter updates are undone; however unlike the typical GBML algorithm, the loss is computed again with the updated codebook vectors, and one step of gradient descent is performed to update the encoder / decoder parameters [keeping the codebook now fixed]. This process is repeated until convergence.

Intuitively, this approach tries to find a good set of codebook vectors by looking-ahead to see how the Encoders'/Decoders' parameters will drift. The authors

**Strengths:**

I really enjoyed reading this paper. I see its strengths as:

1. Very well written and easy to understand. A reader with familiarity of gradient-based meta-learning can read a couple paragraphs in the introduction, look at Figure 1, and immediately understand the proposed approach.
2. Comparison to Huh et al. 2023: I was looking for this comparison after reading the introduction, and I appreciate the thorough discourse comparing to a very close method.
3. Finite difference approximation: this isn't new, but it's quite the nice touch to make differentiating through multiple encoder/decoder updates practical.
4. Section 4.3 is also quite good: I've seen very few mentions of the *indirect* gradients that appear during meta-learning (or other transfer learning techniques) that appear when you differentiate through a prior parameter update.

**Weaknesses:**

This paper also presents several weaknesses:

0. Novelty: this paper applies an existing technique (gradient-based meta-learning) to a new setting (VQ-VAE training). That's not an issue in itself -- the ViT paper was incredible even though it applied an existing technique (attention) to a new setting (computer vision). But ViTs were very practical to train and seemed to work really well on computer vision benchmarks. This leads me to my second point:

1. I don't think this method is practical; it is difficult to imagine a researcher or engineer implementing it as they chase sota performance. Implementing a bi-level optimization loop is quite difficult, and many things can go wrong. Even in the meta-learning community, we try to stay away from this as much as possible. Running a bi-level optimization to train a VQ-VAE would be very difficult for real applications. Even with new frameworks like Betty that make it a bit easier, my understanding is that still very difficult and popular training settings like distributed data parallel would not be supported.

 2. The contribution of Huh et al. 2023 in last years ICLR weakens this paper's contribution. I found many of the ideas in that work highly engaging, and while this work advances them in the direction of meta-learning, my sense is that the results / ideas in this paper are less new in the backdrop of Huh et al. 2023.

3. The authors highlight the significance of indirect gradients in red throughout their paper (especially Figure 1 and Section 4.3). When I looked at these for a different application (in meta-learning), it seemed as their gradient norms were **many orders of magnitude less** then the direct gradients. In this context, my prior is that $||\frac{\partial \mathcal{L'}}{\partial \mathcal{C}}|| >>> ||\frac{\partial \phi'}{\partial \mathcal{C}} \frac{\partial \mathcal{L}'}{\partial \phi '}||$ and $||\frac{\partial \mathcal{L'}}{\partial \mathcal{C}}|| >>> ||\frac{\partial \theta'}{\partial \mathcal{C}} \frac{\partial \mathcal{L}'}{\partial \theta '}||$, therefore the effect of the indirect gradients would be almost nothing. I think it is a weakness to not have explored the dynamics of these effects.

4. The experimental results are limited and evaluated on non-realistic datasets. I'm very empathetic to limited computational resources, but evaluating on two small datasets (FashionMNIST and CIFAR100) with a 2017-style VQ-VAE does not convey to me that this would work for a more recent VQ-VAE architecture (such as a VQGAN).

5. The experimental results section seem inconsistent with Huh et al. 2023 -- specifically Table 3 where VQ-VAE + alt + sync is strictly worse than the default VQ-VAE? Is the training setting unstable, or are you reporting that these two techniques actually result in worse performance than the default model for FashionMNIST?

6. The claim that "MQ-VAE can learn a superior codebook" seems unsupported by the analysis and should be contextualized in the backdrop of recent methods that find not learning the codebook at all can substantively improve performance [1, 2].

[1] Finite Scalar Quantization: VQ-VAE Made Simple, Mentzer et al., ICLR 2024

[2] Language Model Beats Diffusion - Tokenizer is key to visual generation, Yu et al., ICLR 2024

**Questions:**

Please see weaknesses. In particular, addressing:

1. The practicality of this method when implementing a very strong VQ-VAE, and if you believe the results you present on the datasets and architectures that you've chosen provide sufficient evidence to support this claim: "We demonstrate the superiority of MQ-VAE with two computer vision tasks. MQ-VAE significantly outperforms several comparison baselines and ablation methods, highlighting its effectiveness."

2. Inconsistency between Huh et al. 2023 and the results reported on FashionMNIST.

3. Comparison of the magnitude of indirect gradients to the direct ones. If you 0-out the indirect gradients, will the model still perform well? This study would support your assertions that these indirect gradients are crucial for the performance of your approach.

4. The importance of learning a better codebook when recent work has shown that not learning the codebook is actually quite good.

---

> ### Author Response · Authors · 2024-11-25
>
> We appreciate your constructive feedback very much. We provide our response to your review as follows.
>
> ### Weaknesses
>
> - **Weakness 0**
>
>   Thank you for bringing up this point. We acknowledge that our method does not introduce extensive novelty to the network architecture or the bi-level optimization process. However, to our knowledge, MQ-VAE represents the first step in applying meta-learning techniques to the VQ-VAE setting, which deserves recognition as a contribution.
>
> - **Weakness 1**
>
>   Fortunately, the Betty library is quite advanced and includes many common features such as distributed training, mixed-precision, and gradient accumulation. Please refer to Appendix F in [1] and their documentation (https://leopard-ai.github.io/betty/) for all supported features. We also conducted experiments in larger-scale experimental settings, where we leveraged distributed training. The results confirm the scalability of our method and its usefulness in various datasets. Please see the general response and Appendix G for details.
>
> - **Weakness 2**
>
>   We want to clarify that although our method can be viewed as incremental work building upon [2] by adding an "indirect hyper-gradient" path, the novelty is better understood from a holistic meta-learning perspective. Additionally, deeper analyses of the hyper-gradient, both theoretical (Section 4.4) and empirical (Appendix H), reveal the differences between our method and existing methods. These analyses further support the rationale behind why MQ-VAE performs better.
>
> - **Weakness 3**
>
>   Thank you for raising this valuable suggestion! We have added empirical studies on the dynamics of the gradient paths in terms of norm and cosine similarity. The results are presented in Appendix H, which reveal that the indirect hyper-gradients make a non-negligible contribution to codebook training. Regarding your suggestion to zero out the indirect hyper-gradient and observe the results, we have shown in Section 4.4 that removing the indirect hyper-gradient (i.e., using only the direct hyper-gradient) achieves the same effectiveness as the synchronous update rule proposed in [2]. An ablation study comparing these methods (see Table 3) demonstrates that our method (with indirect hyper-gradient) achieves superior performance compared to the synchronous update rule (without indirect hyper-gradient).
>
> - **Weakness 4**
>
>   We understand your concern. To this end, we have conducted additional experiments with larger-scale experimental settings. These include using modern VQ architectures - VQ-GAN and perceptual losses, as well as larger benchmark datasets like CelebA and Imagenet, all at a resolution of 256x256. The results are consistent with those presented in the main paper, demonstrating that our method achieves superior performance in larger-scale experimental settings. These results are provided in Appendix G.
>
> - **Weakness 5**
>
>   Thank you for raising this point. After careful inspection, we found that alternating optimization may result in unstable behavior if the learning rates for each part are set inappropriately. We tuned the hyperparameters and employed early stopping when performance began to degrade. The results have been corrected and should now be consistent with those presented in [2]. While the overall performance of the ablation baselines improved after tuning the hyperparameters, MQ-VAE still largely outperforms them, demonstrating the effectiveness of our method. Please refer to Table 3 in the rebuttal version for details.
>
> - **Weakness 6**
>
>   We apologize for not making this point clear. Please refer to "Motivation and Intuition Behind Meta-learning and Hyper-gradient" in general response on an intuition of why our method works better and "Importance of Learning an Explicit Codebook" in general response on the importance of learning a better codebook when recent work has shown that not learning the codebook is actually quite good.
>
> ### Questions
>
> Please see the replies in the weaknesses section above.
>
> [1] Choe, Sang Keun, et al. "Betty: An automatic differentiation library for multilevel optimization." *arXiv preprint arXiv:2207.02849* (2022).
>
> [2] Huh, Minyoung et al. "Straightening out the straight-through estimator: Overcoming optimization challenges in vector quantized networks." *International Conference on Machine Learning*. PMLR, 2023.

---

> > ### Comment · Reviewer_rpsn · 2024-11-27
> >
> > I’ve reviewed both the paper and the authors’ responses.
> >
> > The paper lacks experimental results aligned with more recent work in the field (2020 and later). As both reviewer vWH6 and I noted, demonstrating that MQ-VAE can improve Taming Transformer’s VQGAN to achieve a lower FID score than the one reported in their repository (https://github.com/CompVis/taming-transformers) would significantly bolster the effectiveness of MQ-VAE. A key issue with the VQGAN results provided in the rebuttal is that neither model was trained to convergence. The current results results on CIFAR and Fashion MNIST show relatively high FID scores and are not enough to show the approach works.
> >
> > MQ-VAE introduces considerable complexity, requiring a bi-level optimization loop, familiarity with gradient-based meta-learning, the Betty framework, and extended training times. Demonstrating that this added complexity leads to improved performance on widely used benchmarks would strengthen the paper. Consequently, I am maintaining my score but recommend that the authors conduct these additional benchmarks and resubmit the work to ICML or NeurIPS.
> >
> > **Weakness 0:** While applying meta-learning techniques outside of few-shot learning is an exciting direction, its impact depends heavily on the benefits provided relative to the complexity and additional costs introduced.
> >
> > **Weakness 1:** You are correct, and I appreciate the clarification. Betty does indeed support DDP.
> >
> > **Weakness 2:** I understand your point, but it seems the appropriate baseline in Table 1 should be VQ-VAE + alt + sync, rather than the baseline VQ-VAE model. While these are included as an ablation in Table 3, the conditions in Table 3 do not fully cover those in Table 1.
> >
> > **Weakness 3:** Appendix H presents an interesting analysis that contradicts my prior assumption that the indirect gradient is significantly smaller than the direct gradient based on meta-learning experiments. This ablation adds valuable insight.
> >
> > **Weakness 4:** I understand the constraints of limited computational resources, but training a VQGAN for fewer than 10k steps is inadequate to demonstrate that MQ-VAE improves performance in this scenario.
> >
> > **Weakness 5:** This raises a concern. Does a similar issue affect other reported results? How were hyperparameters chosen for the results in Tables 1 and 3? Consider including this discussion in the Appendix.
> >
> > **Weakness 6:** I’m confused by the following statement in the general response: “For example, FSQ [2] has reduced expressiveness since there are only a finite number of possible codewords. Additionally, hyperparameter tuning can be difficult as the range of codewords needs to be determined manually for each dimension.” Aren’t there always a finite number of possible codewords (determined by the size of the codebook)? FSQ also offers a mapping to codebook size in Table 1, so in practice, the size of each layer (and the number of layers) does not need to be tuned.

---

> > > ### Author Response · Authors · 2024-11-28
> > >
> > > Thank you for your valuable insights and actively engaged with us.
> > >
> > > - **Weakness 0-5**
> > >   Thank you for your valuable insights and for actively engaging with us. We agree that our additional experiments are not yet convincing enough. We will revise the paper based on your suggestions by conducting more robust experimental studies.
> > >
> > > - **Weakness 6**
> > >   We apologize for not making this point clearer earlier.
> > >   - **Limited expressiveness**
> > >    In the context of finite scalar quantization, by "a finite number of possible codewords," we mean that the implicit codebook contains codewords in the form: $C = \\{ (-1, -1, -1), (-1, -1, 0), (-1, -1, 1), \dots, (1, 1, 1) \\}$ where all coordinates are integers. As a result, an embedding will never be quantized into a vector like $(0.5, 0.5, 0.5)$ because it is not part of the predefined, non-learnable codebook. On the other hand, in vector quantization, it’s possible for an embedding to be quantized into any 3-D vector as long as one of the codewords in the codebook moves there. This is possible because the codebook can be learned, which allows for more flexibility.
> > >
> > >   - **Hyperparameter tuning**
> > >     We agree that this may not be a serious limitation for FSQ, considering  the authors have already provided recommended codebook configurations. However, if a codebook size beyond the provided table is desired, the FSQ levels will need to be manually determined.
> > >
> > > We appreciate the time and effort you’ve put into providing such detailed feedback. Your thoughtful reviews will be invaluable in improving our work and advancing this research area. Thank you once again.

---

### Official Review · Reviewer_ERCU · 2024-10-31

**Soundness:** 3
**Presentation:** 3
**Contribution:** 2
**Rating:** 6
**Confidence:** 3

**Summary:**

This submission proposed to conduct quantization for hidden emebeddings / vectors / codebook in VAE (Variational Auto-Encoder). Specially, it designed a bi-level optimization framework to update encoder/decoder (full-precision parameters) and quantized hidden embeddings alternatively: Normal gradient descent is used in encoder/decoder optimization while the codebook is optimized using a hyper-gradient.

**Strengths:**

- The method proposed is straightforward and writing is easy to understand.

**Weaknesses:**

- My main concern of the submission is the motivation and necessity of hyper-gradient used in codebook optimization. Though author claimed that traditional optimization with STE is task-unaware, the reason behind the claim is not explained. STE is widely used in quantization training while I have not seen it is challenged for unable of catching task information.
- Besides, hyper-gradient is proposed to solve the problem in the bi-level optimization. My understanding towards the problem lie at: the update of lower/upper part is based on an imperfect update of the counterpart. Hyper-gradient is able to perceive the interaction between the two levels. My question is: why hyper-gradient is used in update of quantized codebook instead of encoder/decoder, or why it is not used in both side?

**Questions:**

- To prove the necessity of hyper-gradient, author should provide analytical and empirical experiments. Which baseline in Section 5 is using normal gradient descent in both level?
- Please clarify the motivation of hyper-gradient.

**Details Of Ethics Concerns:**

N.A.

---

> ### Author Response · Authors · 2024-11-25
>
> We appreciate your constructive feedback very much. We provide our response to your review as follows.
>
> ### Weaknesses
>
> - **Weakness 1**
>
>   The STE has several limitations that deserve further improvements, with a number of works already attempting to address this issue (See also the related work section). The STE cannot fully capture task-specific information because, in the original VQ-VAE paper, the objective of training the codebook only blindly focuses on matching the distribution of embeddings and codewords. The task loss (e.g., the reconstruction loss such as MSE) is not explicitly incorporated into the codebook loss. As a result, the gradients for the codebook are not sufficiently informative. Only limited work [1] has attempted to address this issue by introducing the synchronized commitment loss. Our method starts from a meta-learning perspective and can be shown to incorporate the task information by inspecting the hyper-gradient. Moreover, we demonstrated that the synchronized commitment loss [1] has the same effect as the direct hyper-gradient term in our approach.
>
> - **Weakness 2**
>
>   We agree with this point that there are many design choices for applying hyper-gradient descent in this context. Our design choice is motivated by several factors.
>
>   1. In existing literature, the upper layer typically contains far fewer parameters than the lower layer. Relevant examples include neural architecture search [2] and hyperparameter optimization [3].
>   2. Additionally, during VQ-VAE training, the encoder-decoder pair can often be optimized more smoothly due to the sparse gradient characteristics of codebook training. By enabling the codebook to "see into the future," we provide it with additional information, facilitating a more balanced interaction between the two components and allowing the codebook to also account for the task loss. Applying hyper-gradient descent on an encoder-decoder pair does not enjoy such a motivation.
>   3. When hyper-gradient is applied to the codebook, it shows a strong connection to existing work after a close inspection, which further justifies this design choice. We did not find an obvious connection when the hyper-gradient is applied to the other side.
>
>   The other two design choices are also possible, but we did not find them to be strongly motivated to improve VQ-VAE. Nevertheless, we have included empirical studies comparing these design choices in Appendix E. The results show that using hyper-gradient descent for the codebook does not result in performance improvements comparable to MQ-VAE. While using hyper-gradient descent for both sides achieves slightly improved performance, it is less computationally efficient.
>
> ### Questions
>
> - **Question 1**
>
>   For a theoretical analysis, please refer to Sections 4.3 and 4.4. For an empirical analysis, please see Appendix H. Please also refer to the general response for more details.
>
>   In Section 5, except MQ-VAE and methods involving "+MQ-VAE" apply hyper-gradient on the upper level, all other baselines use normal gradient descent in both levels. To our knowledge, our work is the first to introduce a meta-learning-like method to VQ-VAE, and other works mainly focus on improving codebook structure or loss design, and they do not involve the concept of hyper-gradient.
>
> - **Question 2**
>
>   We apologize for not making this point clear. Please refer to "Motivation and Intuition Behind Meta-learning and Hyper-gradient" in the general response for details.
>
> [1] Huh, Minyoung, et al. "Straightening out the straight-through estimator: Overcoming optimization challenges in vector quantized networks." *International Conference on Machine Learning*. PMLR, 2023.
>
> [2] Liu, Hanxiao, Karen Simonyan, and Yiming Yang. "Darts: Differentiable architecture search." *arXiv preprint arXiv:1806.09055* (2018).
>
> [3] Lorraine, Jonathan, Paul Vicol, and David Duvenaud. "Optimizing millions of hyperparameters by implicit differentiation." *International conference on artificial intelligence and statistics*. PMLR, 2020.

---

> > ### Comment · Reviewer_ERCU · 2024-12-02
> > **Raise my score to 6**
> >
> > Though I am not fully convinced by author's explanation on necessity of hyper-gradient, I raise my score to 6 for empirical result demonstrate the effectiveness of hyper-gradient.

---

### Official Review · Reviewer_fShD · 2024-11-02

**Soundness:** 3
**Presentation:** 3
**Contribution:** 3
**Rating:** 6
**Confidence:** 3

**Summary:**

This paper proposes a new training algorithm for learning vector-quantized variational autoencoders (VQ-VAEs). The proposed method interprets the training of VQ-VAE as a bi-level optimization and applies a meta-learning-inspired technique to solve the optimization. The idea seems novel and straightforward, and the experimental results look promising.

**Strengths:**

* The bi-level optimization view of VQ-VAE training is persuasive and seems novel.
* The proposed algorithm is straightforward and achieves strong performance.
* The related works and the background information are carefully covered.

**Weaknesses:**

* An obvious drawback of the proposed method is additional compute overhead and implementation complexity. However, this point is honestly described in Section 5.5 and Table 4. The improvement in various task performance metrics justifies the additional overhead of the algorithm.
* The scale and the diversity of experiments are relatively small compared to modern standards. Including a larger-scale image dataset (e.g., image of size 64x64 or larger) or experiments on a non-image dataset (e.g., medical, audio, time-series, etc.) would greatly enrich the paper.

**Questions:**

* FID evaluation protocol is known to be highly sensitive to implementation details. It would be beneficial in terms of reproducibility to state the detailed procedure involved in FID computation, such as which software package is used and how images are preprocessed.
* Could you include some examples of generated images in the supplementary material so readers can qualitatively appreciate the improvement in generation quality?

---

> ### Author Response · Authors · 2024-11-25
>
> We appreciate your constructive feedback very much. We provide our response to your review as follows.
>
> ### Weaknesses
>
> Thank you for raising the point of limited experiment diversity! To address this, we have conducted additional experiments with larger-scale experimental settings. These include using modern VQ architectures—VQ-GAN and perceptual losses—as well as larger benchmark datasets like CelebA and ImageNet, all at a resolution of 256×256. The results are consistent with those presented in the main paper, demonstrating that our method achieves superior performance in larger-scale experimental settings. These results are provided in Appendix G.
>
> ### Questions
>
> - **Question 1**
>
>   All the metrics for image quality measurement are computed using the torch-metrics library (https://lightning.ai/docs/torchmetrics/stable/). The data preprocessing procedures are simple and straightforward: we convert the image into a tensor with a value range of [0, 1]. For the ImageNet dataset, where the images are not square, we crop the central 256×256 pixels. We have added details about these points to Appendix I.
>
> - **Question 2**
>
>   We added sample reconstructed images evaluated on the Imagenet dataset in Appendix G. The results qualitatively reveal that our method can reconstruct images with higher quality.

---

> > ### Comment · Reviewer_fShD · 2024-11-26
> > **Thanks for the response**
> >
> > Dear authors,
> >
> > I appreciate your comment. Your response addresses my questions, and I would like to keep my evaluation.

---

### Official Review · Reviewer_vWH6 · 2024-11-03

**Soundness:** 2
**Presentation:** 3
**Contribution:** 2
**Rating:** 3
**Confidence:** 4

**Summary:**

The paper proposes to use meta learning to improve training of VQ-VAEs, which are hard to optimize due to the non-differentiability of the vector quantization operation. More specifically the authors propose to optimize the encoder and decoder parameters in the inner loop, and the codebook in the outer loop. The advantage of this approach is that the codebook gets direct feedback from the reconstruction loss, and not indirect feedback as in the original formulation. Different variants of the proposed meta learning algorithm and baselines are compared on CIFAR100 and FashionMNIST reconstruction and image generation. The authors further present results for applying their method to the feature maps of a ResNet.

**Strengths:**

To my knowledge, the application of meta learning to training VQ-VAEs is new. The proposed algorithm is clearly explained, and the paper also proposes some sound tweaks of the algorithm. At a high level, the experiment design makes sense.

**Weaknesses:**

- The experimental results on image reconstruction/generation are not very convincing. The method is only applied to very low resolution, small data sets, and the tasks are quite simple (CIFAR100, FashionMNIST), so it’s unclear how and whether the method generalizes to more modern VQ-VAE designs involving perceptual losses such as VQGAN [1].
- I appreciate that the authors address computational cost in Section 5.5, showing that MQ-VAE requires roughly 4x the compute of the baseline. However, the fair comparison would be to train VQ-VAE 4x longer, but MQ-VAE and VQ-VAE are only compared when trained for the same number of iterations (e.g. Figure 4). In particular it seems that the cheaper alternating optimization and/or synced codebook updates come close to the proposed method (Table 2), so they might match or outperform it in a compute matched comparison.
- It is somewhat unclear whether the FID and IS metrics provided in Table 1 and 3 are reconstruction or generation metrics. Generally the text seems to make not always a clear distinction between the two.
- Some of the methods are “+group” variants. This is not explained as far as I could see, nor is a reference provided.

\
Minor:
- The authors provided code, which is great. Nevertheless, it would be good to provide some details about the model architectures (both VAE and PixelCNN) and optimizers in the paper.

\
[1] Esser et al. "Taming transformers for high-resolution image synthesis." CVPR 2021.

**Questions:**

- All the models have their codebook initialized with k-means. This is not novel, but is usually not required for VQ-VAEs to work reasonably well. Did the authors observe optimization issues when randomly initializing the codebook?
- Some recent works [2, 3] do not use an explicit codebook, but rather just scalar-quantize feature maps. These methods do not suffer from a “task-unaware codebook” since all parameters get gradients from the task loss. Could the authors comment on this?

\
[2] Yu et al. "Language Model Beats Diffusion--Tokenizer is Key to Visual Generation." ICLR 2024
[3] Mentzer et al. "Finite scalar quantization: Vq-vae made simple." ICLR 2024

---

> ### Author Response · Authors · 2024-11-25
>
> We appreciate your constructive feedback very much. We provide our response to your review as follows.
>
> ### Weaknesses
>
> - **Weakness 1**
>
>   We thank all reviewers for raising concerns about the insufficient experimental evaluation. To address this, we have conducted additional experiments with larger-scale experimental settings. These include using modern VQ architectures - VQ-GAN and perceptual losses, as well as larger benchmark datasets like CelebA and Imagenet, all at a resolution of 256x256. The results are consistent with those presented in the main paper, demonstrating that our method achieves superior performance in larger-scale experimental settings. These results are provided in Appendix G.
>
> - **Weakness 2**
>
>   We acknowledge that compared to other improved VQ-VAE methods, our method may not achieve higher performance within a compute-matched comparison. However, in all our experiments, we set the number of iterations sufficiently high to ensure that all compared methods converged to their optimum. This means if a sufficient time budget is possible where all methods converge, our method can achieve better optimal performance compared to ablation baselines.
>
> - **Weakness 3**
>
>   Thank you for bringing up this point! The metrics used in Table 2 are all quality measures of the reconstructed images. We have added clearer explanations in the table captions.
>
> - **Weakness 4**
>
>   The meaning of the "+group" variant is presented at the beginning of Section 5, denoting a series of VQ-VAE variants, including [1-3], where the codeword's dimensionality is split into several groups and quantization performed in each group separately. Please refer to the related work section and the references therein for more details.
>
> - **Minor Weakness**
>
>   We have added additional implementation details in Appendix I, including the model architecture and optimizer, which are largely adapted from existing works to ensure consistency.
>
> ### Questions
>
> - **Question 1**
>
>   In our preliminary experiments, we found that K-means initialization is necessary for a robust startup of codebook training. The issue is that a randomly initialized codebook may lie outside the distribution of image embeddings, meaning those outlier codewords will never be selected. It depends on the input range and the initialization of network weights and may worsen the index collapse issue. Data-dependent initialization, such as K-means, is a common practice to mitigate this issue. In our experiments, K-means initialization was used for all baselines to ensure a fair comparison. For a detailed explanation of K-means initialization in VQ-VAE, please refer to [4].
>
> - **Question 2**
>
>   We are aware that some existing works have considered implicit codebooks. However, these methods belong to a different research branch and have several limitations compared to explicit codebook-based methods. For example, FSQ [5] has reduced expressiveness since there are only a finite number of possible codewords. Additionally, hyperparameter tuning can be difficult as the range of codewords needs to be determined manually for each dimension. These limitations underscore the need for advances in explicit codebook-based methods, and our work aims to address this.
>
>
>
> [1] Kaiser, Lukasz, et al. "Fast decoding in sequence models using discrete latent variables." *International Conference on Machine Learning*. PMLR, 2018.
>
> [2] Guo, Haohan, et al. "Addressing Index Collapse of Large-Codebook Speech Tokenizer with Dual-Decoding Product-Quantized Variational Auto-Encoder." *arXiv preprint arXiv:2406.02940* (2024).
>
> [3] Yu, Jiahui, et al. "Vector-quantized image modeling with improved VQGAN." *arXiv preprint arXiv:2110.04627* (2021).
>
> [4] Łańcucki, Adrian, et al. "Robust training of vector quantized bottleneck models." *2020 International Joint Conference on Neural Networks (IJCNN)*. IEEE, 2020.
>
> [5] Mentzer, Fabian, et al. "Finite Scalar Quantization: VQ-VAE Made Simple." *The Twelfth International Conference on Learning Representations*.

---

> > ### Comment · Reviewer_vWH6 · 2024-11-27
> >
> > I thank the authors for their detailed response and additional experiments. I acknowledge that quite some work was invested. Unfortunately, most of the responses and experiments do not convince me. To be more specific:
> > * The VQGAN FID very high, so I think it's questionable as a baseline. VQ-VAE2 (Razavi et al. 2019) obtained a lower FID about 5 years ago, without using any perceptual losses to train the VQ-VAE.
> > * I don't think VQGAN uses k-means initialization. At least I could not find it documented in the paper or open-source code.
> > * I don't follow the argument about implicit codebooks, that associated models have "reduced expressiveness since there are only a finite number of possible codewords". The purpose of VQ is to have a finite number of codewords; if this was not a requirement one would not have to deal with the challenges associated with VQ.

---

> > > ### Author Response · Authors · 2024-11-28
> > >
> > > Thank you for your valuable insights and actively engaged with us. We would like to clarify further and address your concern.
> > >
> > > > The VQGAN FID is very high, so I think it's questionable as a baseline.
> > >
> > > We agree our additional experiments are not convincing enough for now. Due to time limitations, we only trained all methods for 10k iterations, which may not reach the original performance as stated in their paper. We will revise our paper according to your suggestions by conducting more solid experimental studies.
> > >
> > > > I don't think VQGAN uses k-means initialization.
> > >
> > > While the original VQGAN method did not employ a K-means initialization, it has been shown to be useful for improving the codebook utilization. Please refer to one recent work [1], where the authors used K-means clustering on feature embeddings to initialize the codebook.
> > >
> > > > I don't follow the argument about implicit codebooks.
> > >
> > > We apologize for not making this point clear. In finite scalar quantization, what we mean by "a finite number of possible codewords" is that the implicit codebook contains codebook in the form like $C = \\{(−1, −1, −1),(−1, −1, 0),(−1, −1, 1), . . . ,(1, 1, 1)\\}$ where all coordinate are integers. Therefore, an embedding will never be quantized in a vector like $(0.5, 0.5, 0.5)$, which is not in the set of the predefined, no-learnable codebook. On the other hand, in vector quantization, it is possible that an embedding can be quantized into any 3-D vector as long as one codeword in the codebook moves there. Since the codebook can be learned, this could happen.
> > >
> > > We appreciate your time and detailed feedback, which will be valuable for improving our work.  Thank you for your efforts in making this research area better through your thoughtful reviews.
> > >
> > >
> > >
> > > [1] Zhu, Lei, et al. "Scaling the Codebook Size of VQGAN to 100,000 with a Utilization Rate of 99%." *arXiv preprint arXiv:2406.11837* (2024).

---

### Author Response · Authors · 2024-11-25
**General Response**

Dear reviewers,

We highly value your concerns, and we have submitted the rebuttal version of the paper with major changes highlighted in purple. Based on the valuable feedback from the reviewers, the main concerns regarding MQ-VAE are related to the lack of experiments, clarity on the motivation for using hyper-gradient, and the effectiveness of the hyper-gradient technique. We have made the following efforts to improve our work.

### Additional Experiments with Scaled-up Experimental Settings

We thank all reviewers for raising concerns about the insufficient experimental evaluation. To address your concern, we have conducted additional experiments with larger-scale experimental settings. These include using modern VQ architectures - VQ-GAN and perceptual losses, as well as larger benchmark datasets like CelebA and Imagenet, all at a resolution of 256x256. The results are consistent with those presented in the main paper, demonstrating that our method achieves superior performance in larger-scale experimental settings. These results are provided in Appendix G.

### Motivation and Intuition Behind Meta-learning and Hyper-gradient

Several reviewers raised concerns about the motivation for applying hyper-gradient and questioned why it helps in learning a better codebook.

**We want to clarify that a more intuitive understanding of our method should be motivated from a holistic meta-learning perspective.** In the spirit of meta-learning, MQ-VAE treats the codebook as hyperparameters, with the objective of improving the downstream training of the encoder and decoder. The effectiveness of the codebook is validated and updated based on the performance potential of the encoder and decoder, analogous to how meta-learning evaluates the effectiveness of model initialization for fast network adaptation [1]. The concept of hyper-gradient emerges when we consider gradient-based meta-learning [1], where the bilevel optimization problem transforms into an equivalent formulation. In this case, a hyper-gradient is used for the upper level (i.e., the codebook).

**We are then further motivated to analyze why the gradient-based meta-learning and the hyper-gradient are particularly effective in our setting.** By carefully examining the hyper-gradient, we show that it addresses three challenges specific to our setting, as outlined in the introduction. We also provide preliminary results of gradient analysis and its connection to existing work in Section 4.3 and Section 4.4. We also provide empirical analysis in Appendix H. We acknowledge that the effectiveness of hyper-gradient may not be fully intuitive when examined by parts. A more in-depth and solid analysis, such as a generalization bound, of each part of the hyper-gradient could be a valuable future direction.

### 3. Empirical Analysis of Hyper-gradient

In our rebuttal version, we also added empirical studies on the hyper-gradient in Appendix H. Specifically, we examined the dynamics of both direct and indirect gradients. Our main finding is that the indirect gradient is non-negligible in terms of its norm. Additionally, the direct and indirect gradients are not highly correlated in terms of cosine similarity, indicating that the hyper-gradient can enrich the gradient guidance for the codebook. For further details, please refer to Appendix H.

### 4. Importance of Learning an Explicit Codebook

Two reviewers (reviewer rpsn and reviewer vWH6) raised questions about methods using implicit codebooks [2, 3], which do not have trainable parameters in the quantization layer but generally perform well. However, these methods belong to a different research branch and have several limitations compared to explicit codebook-based methods. For example, FSQ [2] has reduced expressiveness since there are only a finite number of possible codewords. Additionally, hyperparameter tuning can be difficult as the range of codewords needs to be determined manually for each dimension. These limitations underscore the need for advances in explicit codebook-based methods, and our work aims to address this.

[1] Finn, Chelsea, Pieter Abbeel, and Sergey Levine. "Model-agnostic meta-learning for fast adaptation of deep networks." *International Conference on Machine Learning*. PMLR, 2017.

[2] Mentzer, Fabian, et al. "Finite Scalar Quantization: VQ-VAE Made Simple." *The Twelfth International Conference on Learning Representations*.

[3] Yu, Lijun, et al. "Language Model Beats Diffusion-Tokenizer is key to visual generation." *The Twelfth International Conference on Learning Representations*.

---

> ### Author Response · Authors · 2024-11-25
> **General Response (Cont)**
>
> ### 5. Other Updates
>
> 1. We provide a comparison of three design choices for applying hyper-gradient descent in Appendix E, supported by empirical studies. The results show that applying hyper-gradient to the codebook yields better performance than applying to the encoder-decoder pair. Besides, applying hyper-gradient to both sides is slightly more performant but more computationally costly.
>
> 2. Implementation details are provided in Appendix I for reproducibility.
>
> We are open to further discussion regarding the rebuttal version. If you still have any questions, please do not hesitate to share them. Again, thank you all for your valuable reviews of our work.

---

### Author Response · Authors · 2024-11-27
**Additional Experiment Results**

Dear reviewers,

Thank you for your valuable insights and suggestions. We put the our additional experiment results here for your convenience. Please also view the other additional results, such as generated samples (Appendix G) and gradient dynamic (Appendix I), in our rebuttal version.

### Main results in Appendix G

we conducted additional experiments with larger-scale experimental settings. These include using modern VQ architectures - VQ-GAN and perceptual losses, as well as larger benchmark datasets like CelebA and Imagenet, all at a resolution of 256x256. The main results are presented in the following table. We refer to our meta-learned VQ-GAN as MQ-GAN.

| Dataset       | Method | FID ↓    | IS ↑     | LPIPS (10⁻¹) ↓ | Perceptual loss ↓ | Perplexity ↑ |
| ------------- | ------ | -------- | -------- | -------------- | ----------------- | ------------ |
| **CelebA-HQ** | VQ-GAN | 25.1     | 2.90     | 1.34           | 0.242             | 79.5         |
|               | MQ-GAN | **23.2** | **2.97** | **1.28**       | **0.218**         | **79.7**     |
| **Imagenet**  | VQ-GAN | 34.5     | 1.83     | 1.26           | 0.163             | 77.4         |
|               | MQ-GAN | **30.7** | **1.92** | **1.15**       | **0.149**         | **80.8**     |

The results are consistent with those presented in the main paper, demonstrating that our method achieves superior performance in larger-scale experimental settings.

### Main results in Appendix E

We provide a comparison of three design choices for applying hyper-gradient descent in Appendix E. The results are presented below. The methods are evaluated in the CIFAR100 dataset.

| Method                                             | FID ↓    | IS ↑     | LPIPS (10⁻¹) ↓ | MSE (10⁻³) ↓ |
| -------------------------------------------------- | -------- | -------- | -------------- | ------------ |
| VQ-VAE                                             | 81.8     | 4.99     | 1.88           | 7.73         |
| MQ-VAE (Hyper-gradient applied to codebook)        | *53.5*   | **6.40** | *1.25*         | *5.02*       |
| MQ-VAE (Hyper-gradient applied to encoder-decoder) | 56.1     | 5.93     | 1.44           | 5.48         |
| MQ-VAE (Hyper-gradient applied to both sides)      | **52.8** | **6.40** | **1.23**       | **4.58**     |

The results show that applying hyper-gradient to the codebook yields better performance than applying to the encoder-decoder pair. Besides, applying hyper-gradient to both sides is slightly more performant but more computationally costly. This again highlights the effectiveness of our design choice of selecting the codebook as the upper level and applying hyper-gradient to it.

Thank you again for engaging with us and for your valuable feedback.



Best regards,

Authors

---

### Meta-Review · Area_Chair_cPga · 2024-12-20

**Metareview:**

This work introduces a meta-learning solution for training a vector quantized VAE, named MQ-VAE. The authors highlight the shortcomings of vector quantized VAEs, such as code collapse and gradient computation, and formulate the learning process as a bi-level optimization problem. This leads to a formulation where the codebook of the VQ-VAE is learned in the outer-level problem, while the network parameters are learned in the inner-level optimization. Experiments on CIFAR and MNIST demonstrate that MQ-VAE outperforms VQ-VAE comfortably.

The paper was reviewed by four experts in the field. The primary concern raised by the reviewers was the lack of evaluations on large-scale problems, which was only partially addressed during the author-reviewer discussion period. Unfortunately, none of the reviewers championed the paper, as the lack of comprehensive empirical evaluation remains unresolved.

The AC agrees with the reviewers and, sadly, recommends `rejection.` We hope this review helps the authors identify areas for improvement. On a semi-related note, recent advances in bilevel optimization suggest that first-order methods can expedite the optimization process (e.g., Liu et al., *BOME! Bilevel Optimization Made Easy: A Simple First-Order Approach*, NeurIPS'22). Incorporating such methods might make the learning of MQ-VAE faster.

**Additional Comments On Reviewer Discussion:**

The primary concern of the reviewers was the lack of comprehensive large-scale evaluations using the proposed MQ-VAE method. During the author-reviewer discussion period, the authors attempted to address this concern; however, their efforts were not sufficient to convince the reviewers. Additionally, one may assume that the use of a bilevel optimization framework—shown in the paper to be 4x slower on small problems— may contribute to the scalability of the solution to larger problems.  In the post-rebuttal discussion with the AC, the consensus among reviewers was that the paper needs to position itself more effectively for large-scale problems. As a result, none of the reviewers championed the work.  The AC agrees with the reviewers' assessment and, sadly, recommends `rejection.`

---

### Decision · Program_Chairs · 2025-01-22

Reject